

# BBBW on the spindle

Antonio Amariti[1][*], Salvo Mancani[2][†], Davide Morgante[1,3][‡],
Nicolò Petri[4][◦] and Alessia Segati[1,3][§]

**1** INFN, Sezione di Milano, Via Celoria 16, I-20133 Milano, Italy
**2** Physique Théorique et Mathématique and International Solvay Institutes
Université Libre de Bruxelles, C.P.231, 1050 Brussels, Belgium
**3** Dipartimento di Fisica, Università degli studi di Milano,
Via Celoria 16, I-20133, Milano, Italy
**4** Department of Physics, Ben-Gurion University of the Negev,
Be'er-Sheva 84105, Israel

[*] antonio.amariti@mi.infn.it , [†] mancanisalvo@gmail.com , [‡] davide.morgante@mi.infn.it ,
[◦] petri@post.bgu.ac.il , [§] alessia.segati@mi.infn.it

## Abstract

We study the spindle compactification of families of AdS$_5$ consistent truncations corresponding to M5 branes wrapped on complex curves in Calabi-Yau three-folds. From the AdS/CFT correspondence these models are dual to $\mathcal{N} = 1$ SCFTs obtained by gluing of $T_N$ blocks. The truncations considered here have both vector and hyper multiplets and the analysis of the BPS equations on the spindle allows to extract the central charges. Such analysis gives also consistency conditions for the existence of the solutions. The solutions are then found both analytically and numerically for opportune choices of the charges for some sub-families of truncations. We then compare our results with the one expected from the field theory side, by integrating the anomaly polynomial.



# 1 Introduction

A prediction of the AdS/CFT correspondence is the matching of exact quantities of a CFT with their gravitational counterparts. An ancestor result in this direction was obtained in [1], where the central charges of a 2d CFT was computed in terms of an AdS$_3$ gravitational background. Furthermore in absence of a Lagrangian description of an interacting fixed point the correspondence represents a definition of the desired CFT. Another way to produce superconformal field theories consists of compactifying higher dimensional theories on curved manifolds, preserving some supersymmetry by turning on quantized magnetic background fluxes for the global symmetries. Such mechanism, commonly referred to as (partial) topological twist [2–4], has been vastly studied in many stringy and holographic setups.

    The prototypical example was discussed in [5] in terms of branes wrapped on Riemann surfaces. From the gravitational side the mechanism is usually referred as a (gravitational) flow across dimensions. Then in [6] such flows have been generalized and related to the c-extremization principle of [7]. The c-extremization principle in this case is related to a gravitational attractor mechanism (see [8–12] for related works in this direction).

    Recently it has been observed that one can extend the notion of the topological twist on manifolds with orbifold singularities [13]. The explicit orbifold considered in [14] is the spindle, topologically a two sphere with deficit angles at the poles. Supersymmetry in this case is preserved such that the Killing spinors are neither constant nor chiral on the orbifold. Furthermore, there are two ways to preserve supersymmetry, denoted as the twist and the anti-twist. Many field theoretical and gravitational constructions have been proposed in the recent years by considering compactifications on orbifolds [14–39].

    In this paper we will focus on the case of M5 branes wrapped on a complex curve $\mathcal{C}_{\mathfrak{g}}$ in a Calabi-Yau three-fold $X$ [40, 41]. These models are a generalization of the ones obtained in [5] where M5 branes wrapped on a Riemann surface were considered. The construction of [40, 41] generates an infinite family of 4d SCFTs obtained by gluing $T_N$ theories [42]. The setup is specified by two integers that depend on the local geometry of $X$, corresponding to a decomposable $\mathbf{C}^2$ bundle over $\mathcal{C}_{\mathfrak{g}}$. The (non-negative) integers, denoted as $p$ and $q$, are the Chern numbers of the line bundles $\mathcal{L}_{1,2}$ that specify $\mathcal{L}_1 \oplus \mathcal{L}_2 \rightarrow \mathcal{C}_{\mathfrak{g}}$. For $p = q$ the $\mathcal{N} = 1$ case studied in [5] is recovered, while $p = 0$ (or $q = 0$) corresponds to the $\mathcal{N} = 2$ case of [5]. For other choices of $p$ and $q$ the 4d SCFT corresponds to a different $\mathcal{N} = 1$ SCFT.

While M5 branes and the theories of [5] have been already studied on the spindle in various setups [17, 18, 26, 28] a general analysis for the models introduced in [40, 41] has not been pursued so far. Here we are interested in generic choices of $p$ and $q$ from the supergravity perspective. Our starting point are the 5d consistent truncations obtained in full generality by [43] (see also [44–47] for earlier results in this direction). Such truncations have the advantage to hold for any choice of $p$ and $q$, but the price to pay in this case is the presence of hypermultiplets. Anyway, by exploiting the general recipe of [48], we can analyze the reduction on the spindle of the consistent truncations of [43] even in presence of hypermultiplets. The reason is that in this case one hyperscalar triggers an Higgs mechanism that gives a mass to one of the vector multiplets. The Higgsing simplifies the analysis of the BPS equations and of the fluxes at the poles of the spindle, allowing to find the boundary conditions that most of the scalars have to satisfy at the poles in order to compute the central charges in the twist and in the anti-twist class. While this analysis makes the calculation of the central charges possible, it does not guarantee the existence of a solution. Furthermore, it does not fix the boundary condition for the hyperscalar.

However, by restricting to the graviton sector, the universal analytic solution of the type discussed in [13, 15] is found. In this case the scalars are fixed to their $\text{AdS}_5$ value. Observe that the universal twist is consistent only if the 4d superconformal $R$-charge is rational, and this limits the amount of accessible truncations. For more general twists, beyond the universal one, we solved numerically the BPS equations for various values of the hyperscalar at one of the poles of the spindle. When the (unique) value of the hyperscalar that solves the BPS equation, at such pole of the spindle, is found, the existence of the solution is guaranteed. The procedure fixes also the boundary condition for the hyperscalar at the other pole and the finite distance between the poles.

In the following we will exploit such procedure for the consistent truncations of [43] and we will compare our results with the one found on the field theory side by integrating the anomaly polynomial.

The paper is organized as follows. In section 2 we study the spindle compactification of the 4d non lagrangian theories obtained in [41]. First, in sub-section 2.1, we review the relevant aspects of the construction of [41] focusing on the 't Hooft anomalies and on the distinction between the trial $R$-symmetry emerging from the higher dimensional picture and the exact one due to a-maximization. This distinction indeed plays a crucial role in the analysis. Then in sub-section 2.2 we study the compactification on the spindle and we compute the central charge of the emerging two-dimensional theory. In the computation of the exact 2d $R$-symmetry we observe that the result can be formulated (when the conditions of integerness on the fluxes is satisfied) in terms of the 4d trial $R$-symmetry or in terms of the 4d exact one. As a bonus we also study in subsection 2.3 the case of the spindle compactification of 4d models associated to negative degree bundles, corresponding to the models obtained in [49, 50]. The in section 3 we review the supergravity truncation of [43] in order to fix the notations and the conventions that we use in subsequents sections of the paper. In section 4 we study the compactification of the spindle of these 5d $\mathcal{N} = 2$ gauged supergravities, obtaining the relevant BPS and Maxwell equations. In section 5 we focus on the calculation of the conserved charges and of the integer fluxes. In this way we can fix most of the scalars at their boundary values on the spindle and from these results we extract the exact central charges form the gravitational perspective. We eventually observe that these results agree with the ones obtained from the field theoretical analysis. In section 6 we complete our analysis by studying the gravitational solution. First, in sub-section 6.1 we look for an analytical solution, finding that it exists for the universal twist, for choices of $p$ and $q$ that correspond to a rational 4d $R$-symmetry. Then in sub-section 6.2 we look for numerical solutions for more generic values of $p$ and $q$, by turning on also the magnetic charge associated to the flavor symmetry. We find numerical solutions only in

the case of the anti-twist class for Riemann surfaces of positive curvature. In section 7 we conclude by discussing the relation of our results with the literature and by listing a set of open problems not addressed in this paper.

## 2 The 4d SCFT on the spindle

In sub-section 2.1 we are going to review the M-theory construction of $\mathcal{N} = 1$ SCFTs in 4d of [41], which is going to be the starting point for our effective 2d theories compactified on the spindle. These models turn out to be dual to $\mathcal{N} = 1$ SCFT built by opportunely gluing $T_N$ blocks [42]. Then in sub-section 2.2 we construct the theory compactified on the spindle $\Sigma$, closely following [13, 48] mutatis mutandis. Eventually in sub-section 2.3 we study the case of negative degree bundles, obtained in [49, 50], on the spindle.

### 2.1 The 4d model

The worldvolume theory of stack of $N$ M5-branes is well known to be a 6d $\mathcal{N} = (2,0)$ SCFT. One can construct effective 4d theories by wrapping the branes on some specific geometry. In this particular case, we are interested in effective 4d theories obtained by wrapping the M5-branes on a complex Riemann curve of genus $\mathfrak{g}$ $\mathcal{C}_{\mathfrak{g}}$ in a Calabi-Yau three-fold. This geometric construction gives rise to an infinite family of 4d effective theories which are parametrized by two integers depending on the local geometry of the Calabi-Yau three-fold $X$ which in the case of interest is just a holomorphic $\mathbf{C}^2$ bundle over $\mathcal{C}_{\mathfrak{g}}$

$$\mathbf{C}^2 \hookrightarrow X \xrightarrow{\pi} \mathcal{C}_{\mathfrak{g}} . \tag{1}$$

Crucially, when $X$ is decomposable it will take the simpler form $X = \mathcal{L}_1 \oplus \mathcal{L}_2$. This structure has a manifest $U(1)^2$ isometry, one factor for each fiber in the line bundle. The two isometries give rise to two abelian symmetries, one being the $R$-symmetry $U(1)_R$ and the other being an additional flavor symmetry $U(1)_F$.

The integers describing the families of IR $\mathcal{N} = 1$ SCFTs are just the Chern numbers labelling the possible bundle decomposition

$$c_1(\mathcal{L}_1) = p , \qquad c_1(\mathcal{L}_2) = q , \tag{2}$$

subject to the Calabi-Yau condition $p + q = 2(\mathfrak{g} - 1)$. Depending on the choices of these two integers, the fields in the M5-brane theory transform in different representation of the $U(1)_F$ symmetry, leading to different IR fixed points. A solution to the constraint of the Chern numbers is given by the following parametrization

$$p = (1 + \mathbf{z})(\mathfrak{g} - 1) , \qquad q = (1 - \mathbf{z})(\mathfrak{g} - 1) , \tag{3}$$

where $\mathbf{z}(\mathfrak{g} - 1) \in \mathbf{Z}$.

An explicit field theory construction for these theories can be given when the integers $p, q$ in (2) are positive. For these cases in fact the theories can be described, from class-$\mathcal{S}$, as opportune gluing of $2(\mathfrak{g}-1)$ $T_N$ building blocks to create a Riemann surface with no punctures. In the next section we are going also to consider the cases of negative $p, q$ whose explicit construction was given in [49,50] but for which no dual gravity solution is known.

In this setup the key observables are the central charges $c$ and $a$, determined by the following combinations of $R$-symmetry anomalies

$$
\begin{aligned}
c &= \frac{1}{32} \left( 9 \operatorname{Tr} R^3 - 5 \operatorname{Tr} R \right) , \\
a &= \frac{3}{32} \left( 3 \operatorname{Tr} R^3 - \operatorname{Tr} R \right) .
\end{aligned}
\tag{4}
$$

Note that in the large $N$ limit, for holographic SCFTs $a = c$. The central charges can be recovered from the known anomaly polynomial of the M5-brane theory integrated over $\mathcal{C}_\mathfrak{g}$, assuming that no accidental symmetries are generated along the flow. Since the abelian symmetries $U(1)_R$ and $U(1)_F$ mix together, the exact superconformal $R$-symmetry is found by $a$-maximization [51].

One finds that the 't Hooft anomalies of the trial R-charge $R(\epsilon_{4d}) = R + \epsilon_{4d}F$, for theories of type $G = A_N, D_N, E_N$, are given by

$$
\begin{aligned}
\operatorname{Tr} R(\epsilon_{4d})^3 &= (\mathfrak{g}-1)[(r_G + d_G h_G)(1 + \mathbf{z}\epsilon_{4d}^3) - d_G h_G(\epsilon_{4d}^2 + \mathbf{z}\epsilon)], \\
\operatorname{Tr} R(\epsilon_{4d}) &= (\mathfrak{g}-1)r_G(1 + \mathbf{z}\epsilon_{4d}),
\end{aligned}
\tag{5}
$$

where $r_G$, $d_G$ and $h_G$ are the rank, dimension and Coxeter number of $G$ respectively, while $\epsilon$ is the mixing parameter.

We are interested in the $A_{N-1}$ case at large $N$. By plugging (5) into (4) we can use $a$-maximisation to find the superconformal R-charge. This is given by the mixing $R(\epsilon_{4d}^*) \equiv R^* = R + \epsilon_{4d}^*F$ where the mixing parameter at large $N$, fixed by $a$-maximisation, is given by

$$
\epsilon_{4d}^* = \frac{1 + \mathbf{k}\sqrt{1 + 3\mathbf{z}^2}}{3\mathbf{z}},
\tag{6}
$$

where $\mathbf{k}$ is half of the scalar curvature of $\mathcal{C}_\mathfrak{g}$.[1] Choosing $\mathbf{k} = -1$ for later purposes, the 't Hooft anomalies for the superconformal $R$-symmetry read

$$
\begin{aligned}
k_{R^*R^*R^*} &= \frac{2(\mathfrak{g}-1)}{27\mathbf{z}^2}\left[9\mathbf{z}^2 - 1 + (3\mathbf{z}^2 + 1)^{3/2}\right]N^3, & k_{R^*R^*F} &= 0, \\
k_{R^*FF} &= -\frac{(\mathfrak{g}-1)}{3}\sqrt{3\mathbf{z}^2 + 1}N^3, & k_{FFF} &= (\mathfrak{g}-1)\mathbf{z}N^3.
\end{aligned}
\tag{7}
$$

The mixed 't Hooft anomalies between the $R$-symmetry $R$ and the flavor symmetry $F$ can be computed from (5) and they read

$$
\begin{aligned}
k_{RRR} &= (\mathfrak{g}-1)N^3, & k_{RRF} &= -\frac{1}{3}(\mathfrak{g}-1)\mathbf{z}N^3, \\
k_{RFF} &= -\frac{1}{3}(\mathfrak{g}-1)N^3, & k_{FFF} &= (\mathfrak{g}-1)\mathbf{z}N^3.
\end{aligned}
\tag{8}
$$

## 2.2 BBBW on the spindle

Consider the 4d SCFT reviewed above, whose anomaly polynomial in the large $N$ limit reads

$$
I_6 = \frac{1}{6}\sum_{i,j,k=R,F} k_{ijk}\, c_1(F_i)c_1(F_j)c_1(F_k),
\tag{9}
$$

where the coefficients $k_{ijk}$ are given by the mixed 't Hooft anomalies (8) and the $c_1(F_{R,F})$ are the first Chern-classes for the $U(1)$-bundles over the total space $X_4$ with gauge curvature $R$ and $F$.

We proceed to compactify further the 4d theory over the spindle $\Sigma \equiv \mathbf{WCP}^1_{[n_N, n_S]}$, where $n_N, n_S$ label the deficit angles at the north and south pole of the orbifold respectively, with background magnetic fluxes for the two abelian $U(1)_R$ and $U(1)_F$ symmetries of the 4d theory. In order to do that, we need to take into account the azimuthal $U(1)_J$ isometry of the spindle which is generated by rotations about the axis passing through the poles. Geometrically, this is given by considering the total space $X_4$ as a $X_2$ orbibundle fibered over $\Sigma$. In the field theory,

---

[1]The Ricci scalar curvature is normalized such that $\mathbf{k} = 1$ for $\mathfrak{g} = 0$, $\mathbf{k} = 0$ for $\mathfrak{g} = 1$ and $\mathbf{k} = -1$ for $\mathfrak{g} > 1$.

this can be achieved by turning on a connection $A_J$ for the U(1)$_J$ isometry, so that we can write the following gauge connections

$$A^{(I)} = \rho_I(y)(\mathrm{d}z + A_J), \quad I = R, F, \tag{10}$$

where $\rho_I(y)$ are the background fluxes for the abelian symmetries, and $(y, z)$ are respectively the longitundal and azimuthal coordinates over $\Sigma$, with $y \in [y_N, y_S]$ and $z \sim z + 2\pi$. The curvatures for the fields (10) are given by

$$F^{(I)} = \rho_I'(y)\mathrm{d}y \wedge (\mathrm{d}z + A_J) + \rho_I(y)F_J, \quad I = R, F, \tag{11}$$

where $F_J = \mathrm{d}A_J$. These fields are consistent with the flux condition

$$\frac{1}{2\pi} \int_\Sigma F^{(I)} = [\rho_I]_{y_N}^{y_S} = \frac{p_I}{n_S n_N}. \tag{12}$$

The curvature forms $F^{(I)}$ define a U(1)-line bundle $\mathcal{L}_I$ over $X_4$, and the associated first Chern classes are[2]

$$c_1(\mathcal{L}_I) \equiv \left[\frac{F^{(I)}}{2\pi}\right] \in \mathrm{H}^2(X_4, \mathbf{R}), \qquad c_1(J) \equiv \left[\frac{F_J}{2\pi}\right] \in \mathrm{H}^2(X_2, \mathbf{R}). \tag{13}$$

To obtain the 2d anomaly polynomial, we make the following substitution

$$c_1(R) \to c_1(R) + \frac{1}{2}c_1(\mathcal{L}_R), \qquad c_1(F) \to c_1(F) + c_1(\mathcal{L}_F), \tag{14}$$

where $c_1(R)$ and $c_1(F)$ are the pull-back of the U(1)$_R$ and U(1)$_F$ bundles over $X_2$ respectively. The choice of normalization is such that the $R$-symmetry generators give charge 1 to the supercharges. Thus, we shift the curvatures in Eq. (11) accordingly, compute the anomaly polynomial in Eq. (9) and integrate it over $\Sigma$. The result is a combination of the four non-zero mixed 't Hooft anomalies given in sec. 2.1. In the following, as a working example we show only the computation for the terms proportional to $k_{RRR}$

$$\int_\Sigma \left(c_1(R) + \frac{1}{2}c_1(\mathcal{L}_R)\right)^3 = \int_\Sigma \left(\frac{3}{2}c_1(R)^2 c_1(\mathcal{L}_R) + \frac{3}{4}c_1(R)c_1(\mathcal{L}_R)^2 + \frac{1}{8}c_1(\mathcal{L}_R)^3\right), \tag{15}$$

where the product of forms is understood. Notice that the $c_1(R)$ does not depend on the spindle, so they can be factorized out of the integral. Let us consider the first term in (15)

$$\int_\Sigma \frac{3}{2}c_1(R)^2 c_1(\mathcal{L}_R) = \frac{3}{2}c_1(R)^2 \int_\Sigma \frac{F^{(R)}}{2\pi} = \frac{3}{2}c_1(R)^2 [\rho_R]_{y_N}^{y_S}. \tag{16}$$

The second term reads

$$\begin{aligned}
\int_\Sigma \frac{3}{4}c_1(R)c_1(\mathcal{L}_R)^2 &= \frac{3}{4}c_1(R) \int_\Sigma \frac{1}{4\pi^2}F^{(R)} \wedge F^{(R)} \\
&= \frac{3}{4}c_1(R) \int_\Sigma \frac{2}{4\pi^2}\rho_R(y)\rho_R'(y)\mathrm{d}y \wedge (\mathrm{d}z + A_J) \wedge F_J \\
&= \frac{3}{4}c_1(R) \int_\Sigma \frac{1}{4\pi^2}\mathrm{d}\rho_R^2 \wedge (\mathrm{d}z \wedge F_J + A_J \wedge F_J) \\
&= \frac{3}{4}c_1(R)c_1(J) \int_\Sigma \frac{1}{2\pi}\mathrm{d}\rho_R^2 \wedge \mathrm{d}z \\
&= \frac{3}{4}c_1(R)c_1(J)[\rho_R^2]_{y_N}^{y_S},
\end{aligned} \tag{17}$$

[2]Note that the gauge curvature of $J$ is only defined on $X_2$. It's Chern class will not contribute in the integral.

where we used the fact that $A_J \wedge F_J$ is just a total derivative and that $F_J$ does not depend on the spindle as stated in (13). In the second to last step we went back from forms to cohomology classes. The last term in (15) evaluates to

$$\frac{1}{8} \int_{\Sigma} c_1(\mathcal{L}_R)^3 = \frac{1}{8} \int_{\Sigma} \frac{1}{(2\pi)^3} F^{(R)} \wedge F^{(R)} \wedge F^{(R)} = \frac{1}{8} c_1(J)^2 [\rho_R^3]_{y_N}^{y_S}. \tag{18}$$

The complete anomaly 4-form of the 2d theory reads

$$
\begin{aligned}
I_4 = &\frac{1}{4} \left( k_{RRR} [\rho_R]_{y_N}^{y_S} + 2k_{RRF} [\rho_F]_{y_N}^{y_S} \right) c_1(R)^2 + \frac{1}{4} \left( k_{RFF} [\rho_R]_{y_N}^{y_S} + 2 k_{FFF} [\rho_F]_{y_N}^{y_S} \right) c_1(F)^2 \\
&+ \frac{1}{48} \left( k_{RRR} [\rho_R^3]_{y_N}^{y_S} + 8 k_{FFF} [\rho_F^3]_{y_N}^{y_S} + 6 k_{RRF} [\rho_F \rho_R^2]_{y_N}^{y_S} + 12 k_{RFF} [\rho_R \rho_F^2]_{y_N}^{y_S} \right) c_1(J)^2 \\
&+ \frac{1}{2} \left( k_{RRF} [\rho_R]_{y_N}^{y_S} + 2 k_{RFF} [\rho_F]_{y_N}^{y_S} \right) c_1(F) c_1(R) \\
&+ \frac{1}{8} \left( k_{RRR} [\rho_R^2]_{y_N}^{y_S} + 4 k_{RRF} [\rho_R \rho_F]_{y_N}^{y_S} + 4 k_{RFF} [\rho_F^2]_{y_N}^{y_S} \right) c_1(J) c_1(R) \\
&+ \frac{1}{8} \left( 4 k_{FFF} [\rho_F^2]_{y_N}^{y_S} + k_{RRF} [\rho_R^2]_{y_N}^{y_S} + 4 k_{RFF} [\rho_R \rho_F]_{y_N}^{y_S} \right) c_1(J) c_1(F).
\end{aligned}
\tag{19}
$$

To compute the exact central charge we allow a mixing between the various U(1) factors $c_1(J) = \epsilon_{2d} \, c_1(R)$ and $c_1(F) = x_{2d} \, c_1(R)$, extremizing the function

$$c_{\text{trial}}^{2d}(\epsilon_{2d}, x_{2d}) = \frac{6 I_4}{c_1(R)^2}. \tag{20}$$

The background magnetic fluxes are fixed to be

$$\int \frac{F^{(R)}}{2\pi} = \frac{p_R}{n_S n_N}, \qquad \int \frac{F^{(F)}}{2\pi} = \frac{p_F}{n_S n_N}, \tag{21}$$

where $p_R, p_F \in \mathbf{Z}$. For the $R$-symmetry, we have two possible choices of fluxes consistent with supersymmetry

$$\rho_R(y_N) = \frac{(-1)^{t_N}}{n_N}, \qquad \rho_R(y_S) = \frac{(-1)^{t_S+1}}{n_S}, \tag{22}$$

where $t_N = 0, 1$, while $t_S$ is fixed by the twisting procedure, namely $t_S = t_N$ for the twist, while $t_S = t_N + 1$ for the anti-twist. For the flavor symmetry, the flux can be fixed to

$$\rho_F(y_N) = \mathbf{z}_0, \qquad \rho_F(y_S) = \frac{p_F}{n_S n_N} + \mathbf{z}_0, \tag{23}$$

where $\mathbf{z}_0$ is an arbitrary constant.

Let us consider the following parametrization of the on-shell central charge

$$c_{\text{trial}}^{2d}(\epsilon_{2d}^*, x_{2d}^*) \equiv c_{2d}^* = \frac{f(n_S, n_N, p_F; \mathbf{z})}{g(n_S, n_N, p_F; \mathbf{z})} (\mathfrak{g} - 1) N^3. \tag{24}$$

In the case of the twist we have

$$
\begin{aligned}
f(n_S, n_N, p_F; \mathbf{z}) = &\left( (n_N + n_S)^2 - 4 p_F^2 \right) \left( 2 \mathbf{z} p_F + (-1)^{t_N} (n_N + n_S) \right) \\
&\times \left( (-1)^{t_N} (n_N + n_S) \left( 16 \mathbf{z} p_F + (\mathbf{z}^2 + 3)(-1)^{t_N} (n_N + n_S) \right) + 4 \left( 3 \mathbf{z}^2 + 1 \right) p_F^2 \right),
\end{aligned}
\tag{25}
$$

$$
\begin{aligned}
g(n_S, n_N, p_F; \mathbf{z}) = &2 n_N n_S \big( 8 p_F^2 \left( -2 n_N n_S + 3 \mathbf{z}^2 (n_S^2 + n_N^2) \right) - 32 \mathbf{z} p_F^3 (-1)^{t_N} (n_N + n_S) \\
&+ 8 \mathbf{z} p_F (-1)^{t_N} (n_N + n_S) \left( 3 n_N^2 - 2 n_N n_S + 3 n_S^2 \right) \\
&- 48 \mathbf{z}^2 p_F^4 + (n_N + n_S)^2 \left( -2 \left( \mathbf{z}^2 + 2 \right) n_N n_S + \left( \mathbf{z}^2 + 4 \right) n_S^2 + \left( \mathbf{z}^2 + 4 \right) n_N^2 \right) \big).
\end{aligned}
$$

The central charge is extremized by the mixing $\epsilon_{2d}^*, x_{2d}^*$ for which we give the exact, albeit quite cumbersome, result

$$\epsilon_{2d}^* = \frac{\varepsilon(n_S, n_N, p_F; \mathbf{z})}{d(n_S, n_N, p_F; \mathbf{z})}, \qquad x_{2d}^* = \frac{\chi(n_S, n_N, p_F; \mathbf{z})}{d(n_S, n_N, p_F; \mathbf{z})} - \mathbf{z}_0 \epsilon_{2d}^*, \qquad (26)$$

where

$$
\begin{aligned}
\varepsilon(n_S, n_N, p_F; \mathbf{z}) = {}& 4 n_N n_S (-1)^{t_N} (n_N - n_S)(2 n_N (-1)^{t_N}(8\mathbf{z} p_F + (\mathbf{z}^2 + 3) n_S (-1)^{t_N}) \\
& + 16 \mathbf{z} p_F n_S (-1)^{t_N} + 4(3\mathbf{z}^2 + 1) p_F^2 + (\mathbf{z}^2 + 3) n_S^2 + (\mathbf{z}^2 + 3) n_N^2), \\
\chi(n_S, n_N, p_F; \mathbf{z}) = {}& -2 n_S^2 \big( 2 (\mathbf{z}^2 - 3) p_F n_N (-1)^{t_N} - 20 \mathbf{z} p_F^2 + 3 \mathbf{z} n_N^2 \big) \\
& - 4 n_S^3 (-1)^{t_N} \big( \mathbf{z} n_N (-1)^{t_N} - 2 p_F \big) \\
& - 4 \mathbf{z} n_S (-1)^{t_N} \big( n_N^2 - 4 p_F^2 \big)\big( 2\mathbf{z} p_F + n_N (-1)^{t_N} \big) \\
& - 16 (\mathbf{z}^2 + 1) p_F^3 n_N (-1)^{t_N} - 4 (\mathbf{z}^2 + 1) p_F n_N^3 (-1)^{t_N} \\
& - 24 \mathbf{z} p_F^2 n_N^2 - 16 \mathbf{z} p_F^4 - \mathbf{z} n_S^4 - \mathbf{z} n_N^4, \\
d(n_S, n_N, p_F; \mathbf{z}) = {}& 24 \mathbf{z}^2 p_F^2 n_S^2 + 4 n_N^3 (-1)^{t_N} \big( 6\mathbf{z} p_F + (-1)^{t_N} n_S \big) \\
& + 2 \mathbf{z} n_N^2 \big( 4 p_F n_S (-1)^{t_N} + 12 \mathbf{z} p_F^2 - \mathbf{z} n_S^2 \big) \\
& + 4 n_N (-1)^{t_N} \big( n_S^2 - 4 p_F^2 \big)\big( 2\mathbf{z} p_F + n_S (-1)^{t_N} \big) \\
& + 24 \mathbf{z} p_F n_S^3 (-1)^{t_N} - 32 \mathbf{z} p_F^3 n_S (-1)^{t_N} \\
& - 48 \mathbf{z}^2 p_F^4 + (\mathbf{z}^2 + 4) n_S^4 + (\mathbf{z}^2 + 4) n_N^4.
\end{aligned}
\qquad (27)
$$

Notice that there is no explicit $\mathbf{z}_0$ dependence in the central charge.

We can check the validity of the result, by considering the $S^2$ limiting case, where $n_S = n_N = 1, p_F = 0$ and comparing with the result of [6]. As expected the two results match.[3] One can see that in this limit the mixing parameter $\epsilon_{2d}^* = 0$. This is to be expected since in this limit the spindle becomes a $\mathbf{P}^1$, and the abelian $U(1)_J$ is enhanced to the $SU(2)$ isometry of the $\mathbf{P}^1$, thus does not mix anymore with the $R$-symmetry.

Instead, for the anti-twist case the on-shell central charge is given by

$$
\begin{aligned}
f(n_S, n_N, p_F; \mathbf{z}) = {}& \big( (n_S - n_N)^2 - 4 p_F^2 \big)\big( 2\mathbf{z} p_F + (-1)^{t_N}(n_N - n_S) \big) \\
& \times \big( (-1)^{t_N}(n_N - n_S)\big(16\mathbf{z} p_F + (\mathbf{z}^2 + 3)(-1)^{t_N}(n_N - n_S)\big) + 4(3\mathbf{z}^2 + 1) p_F^2 \big), \\
g(n_S, n_N, p_F; \mathbf{z}) = {}& 2 n_N n_S \big( 8 p_F^2 (2 n_N n_S + 3\mathbf{z}^2 n_S^2 + 3\mathbf{z}^2 n_N^2) + 32 \mathbf{z} p_F^3 (-1)^{t_N}(n_S - n_N) \\
& - 8 p_F (-1)^{t_N}(n_S - n_N)(3 n_N^2 + 2 n_N n_S + 3 n_S^2) \\
& - 48 \mathbf{z}^2 p_F^4 + (n_S - n_N)^2 \big( 2 (\mathbf{z}^2 + 2) n_N n_S + (\mathbf{z}^2 + 4) n_S^2 + (\mathbf{z}^2 + 4) n_N^2 \big) \big),
\end{aligned}
\qquad (28)
$$

where the extremum, using the same parametrization as in (26), is reached for the following mixing

$$
\begin{aligned}
\varepsilon(n_S, n_N, p_F; \mathbf{z}) = {}& -4 n_N n_S (-1)^{t_N}(n_N + n_S)\big( 2 n_N (-1)^{t_N}\big(8\mathbf{z} p_F - (-1)^{t_N}(\mathbf{z}^2 + 3) n_S\big) \\
& - 16 \mathbf{z} p_F n_S (-1)^{t_N} + 4(3\mathbf{z}^2 + 1) p_F^2 + (\mathbf{z}^2 + 3)(n_S^2 + n_N^2) \big),
\end{aligned}
\qquad (29)
$$

---

[3]From the result of [6], one fixes $\eta_1 = 2(\mathfrak{g} - 1), \eta_2 = -2, \kappa_1 = -1, \kappa_2 = 1, z_1 = z_2 = \mathbf{z}$ to find the matching.

$$\chi(n_S, n_N, p_F; \mathbf{z}) = -2n_S^2\left(2(-1)^{t_N}\left(\mathbf{z}^2-3\right)p_F n_N - 20\mathbf{z}p_F^2 + 3\mathbf{z}n_N^2\right)$$
$$+ 4(-1)^{t_N}n_S^3\left((-1)^{t_N}\mathbf{z}n_N - 2p_F\right)$$
$$+ 4(-1)^{t_N}\mathbf{z}n_S\left(n_N^2 - 4p_F^2\right)\left(2\mathbf{z}p_F + (-1)^{t_N}n_N\right)$$
$$- 16(-1)^{t_N}\left(\mathbf{z}^2+1\right)p_F^3 n_N - 4(-1)^{t_N}\left(\mathbf{z}^2+1\right)p_F n_N^3$$
$$- 24\mathbf{z}p_F^2 n_N^2 - 16\mathbf{z}p_F^4 - \mathbf{z}(n_S^4 + n_N^4),$$
$$d(n_S, n_N, p_F; \mathbf{z}) = 24\mathbf{z}^2 p_F^2 n_S^2 + 4(-1)^{t_N}n_N^3\left(6\mathbf{z}p_F - (-1)^{t_N}n_S\right)$$
$$+ 2\mathbf{z}n_N^2\left(-4(-1)^{t_N}p_F n_S + 12\mathbf{z}p_F^2 - \mathbf{z}n_S^2\right)$$
$$+ 4(-1)^{t_N}n_N\left(n_S^2 - 4p_F^2\right)\left(2\mathbf{z}p_F - (-1)^{t_N}n_S\right)$$
$$- 24(-1)^{t_N}\mathbf{z}p_F n_S^3 + 32(-1)^{t_N}\mathbf{z}p_F^3 n_S$$
$$- 48\mathbf{z}^2 p_F^4 + \left(\mathbf{z}^2+4\right)(n_S^4 + n_N^4).$$

Once again, the on-shell central charge does not depend on $\mathbf{z}_0$ as expected. Notice that the central charge for the anti-twist case is related to the twist one by $n_S \to -n_S$.

The central charge calculated from the $R^*, F$ anomalies (7) instead of $R$, can be computed in the same manner as just described. The two exact central charges will then match as follows

$$c_{2d}^*\left(\epsilon_{2d}^{(1)*}, x_{2d}^{(1)*}; R, F, n_S(-1)^{t_N} + n_N(-1)^{t_S}, p_F\right)$$
$$= c_{2d}^*\left(\epsilon_{2d}^{(2)*}, x_{2d}^{(2)*}; R^*, F, n_S(-1)^{t_N} + n_N(-1)^{t_S}, p_F + \epsilon_{4d}^* \frac{n_S(-1)^{t_N} + n_N(-1)^{t_S}}{2}\right), \quad (30)$$

where $\epsilon_{4d}^*$ is the 4d mixing parameter found in (6) with $\mathbf{k} = -1$, and we specified which symmetries we are considering as well as their fluxes. Namely, the former is obtained from the anomaly polynomial considering the 't Hooft anomalies (8) and their fluxes, while the latter is obtained considering the anomalies (7) and their fluxes are related with the other by a shift.

Observe that the universal twist[4] is consistent only if the exact 4d $R$-symmetry is rational. From the second line in (30) it follows that this choice requires to set the combination $p_F + \epsilon_{4d}^* \frac{n_S(-1)^{t_N} + n_N(-1)^{t_S}}{2}$ to zero. The integerness conditions on $p_F$, $n_S$ and $n_N$ then restrict the allowed values of $p$ and $q$ admitting the universal twist.

## 2.3 Negative degree bundles

Here we further generalize the construction of [40,41] by gluing $2(\mathfrak{g}-1)$ together copies of $T_N^{(m)}$ theories [52]. This construction reproduces the model of [40,41] when $m = 0$ [49,50] and generalizes it for generic $m$. The construction of [40,41] in fact allows only for positive $p, q \geq 0$, while in the construction of [49,50], one can allow also for negative degree bundles. Although these theories have no known supergravity description at this time, we give the field theory calculation for completeness.

The cubic anomalies of the model of [40,41] can be recovered from the ones of the $T_N^{(m)}$ blocks by linear combination of the $U(1)_i$ isometries of the line bundles. Namely, $R = (J_+ + J_-)/2$

---

[4]The universal twist corresponds, on the field theory side, to twist along the exact $R$-symmetry of the 4d theory. This corresponds indeed on the dual side to turn on a background flux only for the graviphoton in the flow across dimensions from AdS$_5$ to AdS$_3$. In our case we have shown indeed that by using the trial $R$-symmetry $U(1)_R$ and the flavor symmetry $U(1)_F$ the exact R-symmetry $U(1)_{R^*}$ realizes the universal twist. For consistency this procedure requires to restrict the allowed values of $p$ and $q$ such that the exact $R$-symmetry is rational as well. We have further commented on this issues in section 6.

and $F = (J_- - J_+)/2$, following the naming convention of [49]. Therefore, in the large-$N$ limit

$$k_{RRR} = \frac{N^3}{2}, \qquad k_{RRF} = -\frac{1}{6}(1 + 2m)N^3,$$
$$k_{RFF} = -\frac{N^3}{6}, \qquad k_{FFF} = \frac{1}{2}(1 + 2m)N^3, \tag{31}$$

where the integer $m$ parametrizes the degree of the line bundles $p = m + 1$ and $q = -m$.

Following the same arguments as before, we can compactify these theories on the spindle and find the central charge of a family of theories parametrized by $m$. By taking the anomaly polynomial constructed from the anomalies (31), we find the following central charge in the case of the twist

$$
\begin{aligned}
f(n_S, n_N, p_F; m) = {}& 2\left(4p_F^2 - (n_N + n_S)^2\right)\left(2(2m+1)p_F + (-1)^{t_N}(n_S + n_N)\right) \\
& \times \Big((-1)^{t_N}(n_N + n_S)\big(4(2m+1)p_F + \big(m^2 + m + 1\big)(-1)^{t_N}(n_N + n_S)\big) \\
& + 4(3m(m+1) + 1)p_F^2\Big),
\end{aligned} \tag{32}
$$

$$
\begin{aligned}
g(n_S, n_N, p_F; m) = {}& n_N n_S\Big((-1)^{t_N}\big(4n_S^3\big(6(2m+1)p_F + (-1)^{t_N}n_N\big) \\
& + 2(-1)^{t_N}(2m+1)n_S^2\big(12(2m+1)p_F^2 \\
& - (-1)^{t_N}n_N\big((-1)^{t_N}(2m+1)n_N - 4p_F\big)\big) \\
& + 4n_S\big(n_N^2 - 4p_F^2\big)\big(2(2m+1)p_F + (-1)^{t_N}n_N\big) \\
& + n_N\big((-1)^{t_N}n_N\big((-1)^{t_N}n_N\big(24(2m+1)p_F + (-1)^{t_N}(4m(m+1)+5)n_N\big) \\
& + 24(2m+1)^2 p_F^2\big) - 32(2m+1)p_F^3\big) + (-1)^{t_N}(4m(m+1)+5)n_S^4\big) \\
& - 48(2m+1)^2 p_F^4\Big),
\end{aligned} \tag{33}
$$

where we used the parametrization (24). The mixing is given by

$$
\begin{aligned}
\varepsilon(n_S, n_N, p_F; m) = {}& 16 n_N n_S(-1)^{t_N}\Big(4(-1)^{t_N}(2m+1)p_F\big(n_N^2 - n_S^2\big) \\
& + 4(3m(m+1)+1)p_F^2(n_N - n_S) \\
& + \big(m^2 + m + 1\big)(n_N - n_S)(n_N + n_S)^2\Big),
\end{aligned}
$$

$$
\begin{aligned}
\chi(n_S, n_N, p_F; m) = {}& -4n_N^3(-1)^{t_N}\big(2\big(2m^2 + 2m + 1\big)p_F + (-1)^{t_N}(2m+1)n_S\big) \\
& - 4n_N(-1)^{t_N}\big(2\big(2m^2 + 2m - 1\big)p_F n_S^2 + 8\big(2m^2 + 2m + 1\big)p_F^3 \\
& - 4(-1)^{t_N}(2m+1)p_F^2 n_S + (-1)^{t_N}(2m+1)n_S^3\big) \\
& - 2(2m+1)n_N^2\big(4(-1)^{t_N}(2m+1)p_F n_S + 12p_F^2 + 3n_S^2\big) \\
& + 32(-1)^{t_N}(2m+1)^2 p_F^3 n_S + 40(2m+1)p_F^2 n_S^2 \\
& - 16(2m+1)p_F^4 + 8(-1)^{t_N}p_F n_S^3 - (2m+1)n_S^4 - (2m+1)n_N^4,
\end{aligned}
$$

$$
\begin{aligned}
d(n_S, n_N, p_F; m) = {}& -32(-1)^{t_N}(2m+1)p_F^3(n_N + n_S) \\
& + 8p_F^2\big(3(2m+1)^2 n_N^2 + 3(2m+1)^2 n_S^2 - 2n_N n_S\big) \\
& + 8(-1)^{t_N}(2m+1)p_F(n_N + n_S)\big(-2n_N n_S + 3n_N^2 + 3n_S^2\big) \\
& - 48(2m+1)^2 p_F^4 + (n_N + n_S)^2\Big(-2(4m(m+1)+3)n_N n_S \\
& + (4m(m+1)+5)(n_N^2 + n_S^2)\Big).
\end{aligned} \tag{34}
$$

$$f(n_S, n_N, p_F; m) = 2\left((n_N - n_S)^2 - 4p_F^2\right)\left(2(2m+1)p_F + (-1)^{t_N}(n_S + n_N)\right)$$
$$\times \left((-1)^{t_N}(n_N - n_S)\left(4(2m+1)p_F + \left(m^2 + m + 1\right)(-1)^{t_N}(n_N - n_S)\right)\right.$$
$$\left. + 4(3m(m+1)+1)p_F^2\right), \tag{35}$$

$$g(n_S, n_N, p_F; m) = -n_N n_S\left((-1)^{t_N}\left(-4n_S^3\left(6(2m+1)p_F + (-1)^{t_N}n_N\right)\right.\right.$$
$$+ 2(-1)^{t_N}(2m+1)n_S^2\left(12(2m+1)p_F^2\right.$$
$$- (-1)^{t_N}n_N\left((-1)^{t_N}(2m+1)n_N - 4p_F\right)\right)$$
$$- 4n_S\left(n_N^2 - 4p_F^2\right)\left(2(2m+1)p_F + (-1)^{t_N}n_N\right) \tag{36}$$
$$+ n_N\left((-1)^{t_N}n_N\left((-1)^{t_N}n_N\left(24(2m+1)p_F + (-1)^{t_N}(4m(m+1)+5)n_N\right)\right.\right.$$
$$\left.\left. + 24(2m+1)^2 p_F^2\right) - 32(2m+1)p_F^3\right) + (-1)^{t_N}(4m(m+1)+5)n_S^4\right)$$
$$- 48(2m+1)^2 p_F^4\Big),$$

where we used the parametrization (24). The mixing is given by

$$\varepsilon(n_S, n_N, p_F; m) = -16n_N n_S(-1)^{t_N}\left(4(-1)^{t_N}(2m+1)p_F\left(n_N^2 - n_S^2\right)\right.$$
$$+ 4(3m(m+1)+1)p_F^2(n_N + n_S)$$
$$\left. + \left(m^2 + m + 1\right)(n_N + n_S)(n_N - n_S)^2\right),$$

$$\chi(n_S, n_N, p_F; m) = -4n_N^3(-1)^{t_N}\left(2\left(2m^2 + 2m + 1\right)p_F - (-1)^{t_N}(2m+1)n_S\right)$$
$$- 4n_N(-1)^{t_N}\left(2\left(2m^2 + 2m - 1\right)p_F n_S^2 + 8\left(2m^2 + 2m + 1\right)p_F^3\right.$$
$$\left. + 4(-1)^{t_N}(2m+1)p_F^2 n_S - (-1)^{t_N}(2m+1)n_S^3\right)$$
$$- 2(2m+1)n_N^2\left(-4(-1)^{t_N}(2m+1)p_F n_S + 12p_F^2 + 3n_S^2\right) \tag{37}$$
$$- 32(-1)^{t_N}(2m+1)^2 p_F^3 n_S + 40(2m+1)p_F^2 n_S^2$$
$$- 16(2m+1)p_F^4 - 8(-1)^{t_N}p_F n_S^3 - (2m+1)n_S^4 - (2m+1)n_N^4,$$

$$d(n_S, n_N, p_F; m) = -32(-1)^{t_N}(2m+1)p_F^3(n_N - n_S)$$
$$+ 8p_F^2\left(3(2m+1)^2 n_N^2 + 3(2m+1)^2 n_S^2 + 2n_N n_S\right)$$
$$+ 8(-1)^{t_N}(2m+1)p_F(n_N - n_S)\left(2n_N n_S + 3n_N^2 + 3n_S^2\right)$$
$$- 48(2m+1)^2 p_F^4 + (n_N - n_S)^2\left(2(4m(m+1)+3)n_N n_S\right.$$
$$\left. + (4m(m+1)+5)(n_N^2 + n_S^2)\right).$$

In the limit of $m \to 0$ one recovers the same result of the compactified model of [40, 41], as expected.

## 3  The 5d supergravity truncation

The five-dimensional supergravity model we are working with is a consistent truncation from eleven-dimensional supergravity studied in [43]. It contains two vector multiplets and one hypermultiplet and it has gauge group $U(1) \times \mathbf{R}$.

As we mentioned before, this truncation generalizes the structure associated with the solutions of [40,41] and it completes the consistent truncation of seven-dimensional $\mathcal{N} = 4$ SO(5) gauged supergravity reduced on a Riemann surface $\mathcal{C}_{\mathfrak{g}}$ analyzed in [44]. There, the 5d model was obtained truncating the 7d supergravity to the $U(1)^2$ sector, corresponding to the Cartan

of SO(5). Besides enclosing the two U(1) gauge fields and the two scalars belonging to the vector multiplets, the bosonic sector of the construction made in [43] also includes all the scalar fields in the hypermultiplet, and furthermore it gives a direct derivation of the gauging. In the following we outline the construction made in [43]. The eleven-dimensional metric is

$$ds_{11}^2 = e^{2\Delta} ds_{\text{AdS}_5}^2 + ds_6^2, \tag{38}$$

which corresponds to a warped product $\text{AdS}_5 \times_\text{w} \mathcal{M}$ with warp factor $e^{2\Delta}\ell^2 = e^{2f_0}\bar{\Delta}^{1/3}$, where $\ell$ is the AdS radius and $\bar{\Delta}$ and $f_0$ are constants. $\mathcal{M}_6$ is a six-dimensional manifold given by a fibration of a squashed-sphere $\mathcal{M}_4$ over the Riemann surface $\mathcal{C}_\mathfrak{g}$ and has metric

$$ds_6^2 = \bar{\Delta}^{1/3} e^{2g_0} ds_{\mathcal{C}_\mathfrak{g}}^2 + \frac{1}{4}\bar{\Delta}^{-2/3} ds_4^2, \tag{39}$$

where $g_0$ is a constant. The Riemann surface has Ricci scalar curvature $\mathbf{k}$ as discussed after formula (6) and the metric on $\mathcal{M}_4$ is

$$ds_4^2 = X_0^{-1} d\mu_0^2 + \sum_{i=1,2} X_i^{-1}\big(d\mu_i^2 + \mu_i^2(d\varphi_i + A^{(i)})^2\big), \tag{40}$$

with

$$\mu_0 = \cos\zeta, \qquad \mu_1 = \sin\zeta\cos\frac{\theta}{2}, \qquad \mu_2 = \sin\zeta\sin\frac{\theta}{2}. \tag{41}$$

The angles $\varphi_1, \varphi_2$ are in $[0, 2\pi]$, while $\zeta, \theta$ are in $[0, \pi]$. $A^{(1)}$ and $A^{(2)}$ gauge two U(1) isometries of the squashed $S^4$. Furthermore,

$$\bar{\Delta} = \sum_{I=0}^2 X_I \mu_I^2, \qquad e^{f_0} = X_0^{-1}, \qquad e^{2g_0} = -\frac{1}{8}\mathbf{k}X_1 X_2\big[(1-\mathbf{z})X_1 + (1+\mathbf{z})X_2\big], \tag{42}$$

where $\mathbf{z}$, that can be read from (3) as

$$\mathbf{z} = \frac{p-q}{p+q}, \tag{43}$$

is a discrete parameter related to the Chern numbers $p$ and $q$ and

$$\begin{aligned}
X_0 &= (X_1 X_2)^{-2}, \\
X_1 X_2^{-1} &= \frac{1+\mathbf{z}}{2\mathbf{z} - \mathbf{k}\sqrt{1+3\mathbf{z}^2}}, \\
X_1^5 &= \frac{1 + 7\mathbf{z} + 7\mathbf{z}^2 + 33\mathbf{z}^3 + \mathbf{k}(1 + 4\mathbf{z} + 19\mathbf{z}^2)\sqrt{1+3\mathbf{z}^2}}{4\mathbf{z}(1-\mathbf{z})^2}.
\end{aligned} \tag{44}$$

There is also a four-form flux, but we address the interested reader to [43] for its explicit form. Notice that the $\mathcal{N} = 1$ and $\mathcal{N} = 2$ twistings studied in [5] can be recovered as special cases from this model: the first one arises from setting $p = q$ (corresponding to $\mathbf{z} = 0$), while the second one from $p = 0$ or $q = 0$ ($\mathbf{z} = \pm 1$).

## 3.1 $\mathcal{N} = 2$ supergravity structure

The reduction described above gives rise to an infinite family of $\mathcal{N} = 2$ gauged supergravity theories in five dimensions. Here we summarize the most salient features of the model and we refer the reader to appendix A of [37] for a short review of 5d $\mathcal{N} = 2$ gauged supergravity.[5]

---

[5]The Lagrangian in (B.10) of [43] that we are using here can be obtained from the one used in [37] by rescaling the gauge fields and the coupling constant as

$$A_{\text{there}}^I = -\sqrt{\frac{3}{2}} A_{\text{here}}^I, \qquad g_{\text{there}} = -\sqrt{\frac{2}{3}} g_{\text{here}}. \tag{45}$$

Focusing on the vector multiplet sector, the two real scalars $\Sigma$ and $\phi$ parametrize the Very Special Real Manifold

$$\mathcal{M}_V = \mathbf{R}_+ \times SO(1,1), \tag{46}$$

that has metric

$$g_{xy} = \begin{pmatrix} \frac{3}{\Sigma^2} & 0 \\ 0 & 1 \end{pmatrix}. \tag{47}$$

The homogeneous coordinates $h^I(\Sigma, \phi)$ (from now on we will omit the explicit dependence of the sections from the two real scalars $\Sigma$ and $\phi$) are given by

$$h^0 = \frac{1}{\Sigma^2}, \qquad h^1 = -\Sigma H^1, \qquad h^2 = -\Sigma H^2, \tag{48}$$

where

$$H^1 = \sinh\phi, \qquad H^2 = \cosh\phi, \tag{49}$$

parametrize the unit hyperboloid $SO(1,1)$, while $\Sigma$ parametrizes $\mathbf{R}^+$. The metric $g_{xy}$ is the pull-back of the metric $a_{IJ}$ in the ambient space, which takes the form

$$a_{IJ} = \frac{2}{3\Sigma^2} \begin{pmatrix} \frac{\Sigma^6}{2} & 0 & 0 \\ 0 & 2(H^1)^2 + 1 & -2H^1H^2 \\ 0 & -2H^1H^2 & 2(H^2)^2 - 1 \end{pmatrix}. \tag{50}$$

The non-zero components of the totally symmetric tensor $C_{IJK}$ are

$$C_{0\bar{I}\bar{J}} = C_{\bar{I}0\bar{J}} = C_{\bar{I}\bar{J}0} = \frac{1}{3}\eta_{\bar{I}\bar{J}}, \qquad \text{for } \bar{I}, \bar{J} = 1, 2, \tag{51}$$

with $\eta = \text{diag}(-1, 1)$.

Moving to the hypermultiplet sector, the quaternionic manifold

$$\mathcal{M}_H = \frac{SU(2,1)}{SU(2) \times U(1)}, \tag{52}$$

is spanned by the scalars $q^X = \{\varphi, \Xi, \theta_1, \theta_2\}$ with line element[6]

$$g_{XY} dq^X dq^Y = -d\varphi^2 - \frac{1}{2}e^{2\varphi}(d\theta_1^2 + d\theta_2^2) - \frac{1}{4}e^{4\varphi}(d\Xi - \theta_1 d\theta_2 + \theta_2 d\theta_1)^2. \tag{53}$$

Only the hypermultiplet sector is gauged and the corresponding Killing vectors $k_I = k_I^X \partial_X$ read

$$k_0 = \partial_\Xi, \qquad k_1 = \mathbf{zk}\partial_\Xi, \qquad k_2 = -\mathbf{k}\partial_\Xi + 2(\theta_2 \partial_{\theta_1} - \theta_1 \partial_{\theta_2}), \tag{54}$$

with associated Killing prepotentials

$$
\begin{aligned}
P_0^r &= \{0, 0, \frac{1}{4}e^{2\varphi}\}, \\
P_1^r &= \{0, 0, \frac{\mathbf{zk}}{4}e^{2\varphi}\}, \\
P_2^r &= \{\sqrt{2}e^\varphi \theta_1, \sqrt{2}e^\varphi \theta_2, -1 + \frac{1}{4}e^{2\varphi}(2\theta_1^2 + 2\theta_2^2 - \mathbf{k})\}.
\end{aligned}
\tag{55}
$$

Thus, the bosonic part of the five-dimensional Lagrangian is

$$e^{-1}\mathscr{L} = \frac{1}{2}R - \frac{1}{\Sigma^2}\partial_\mu\Sigma\partial^\mu\Sigma - \frac{3}{4}a_{\bar{I}\bar{J}}\partial_\mu(\Sigma H^{\bar{I}})\partial^\mu(\Sigma H^{\bar{J}}) - \frac{1}{2}g_{XY}\mathcal{D}_\mu q^X \mathcal{D}^\mu q^Y \tag{56}$$

$$- \frac{\Sigma^4}{12}F_{\mu\nu}^0 F^{0\mu\nu} - \frac{1}{4}a_{\bar{I}\bar{J}}F_{\mu\nu}^{\bar{I}}F^{\bar{J}\mu\nu} - \frac{e^{-1}}{12}\sqrt{\frac{2}{3}}\epsilon^{\mu\nu\rho\sigma\tau}\left(F_{\mu\nu}^1 F^{\mu\nu 1} - F_{\mu\nu}^2 F^{\mu\nu 2}\right)A_\tau^0 - g^2 V,$$

where we recall the notation $\bar{I}, \bar{J} = 1, 2$ and $V$ represents the scalar potential of the theory.

---

[6]We are using a different normalization w.r.t. [43]. This allows us to obtain a simplified version of the hyperino variation, as it was pointed out in [37].

### 3.2 The model

In the remainder of this paper we will work with a further truncation of the 5d supergravity model introduced above, which is obtained by setting

$$\theta_1 = \theta_2 = 0 \,, \tag{57}$$

consistently with the AdS$_5$ vacuum of the model we started from. In this truncation, the Killing vectors (54) simplify to

$$k_0 = \partial_\Xi \,, \qquad k_1 = \mathbf{z}\mathbf{k}\partial_\Xi \,, \qquad k_2 = -\mathbf{k}\partial_\Xi \,. \tag{58}$$

Notice that from (58) we can see that the field $\Xi$ gets charged under the vector $A_\mu^{(0)} + \mathbf{z}\mathbf{k}A_\mu^{(1)} - \mathbf{k}A_\mu^{(2)}$, that becomes massive. Furthermore, only the third SU(2)-components of the Killing prepotentials (55) survive and they reduce to

$$P_0^3 = \frac{1}{4}e^{2\varphi}\,, \qquad P_1^3 = \frac{\mathbf{z}\mathbf{k}}{4}e^{2\varphi}\,, \qquad P_2^3 = -1 - \frac{\mathbf{k}}{4}e^{2\varphi}\,. \tag{59}$$

We can thus introduce a superpotential as

$$W = h^I P_I^3 = \frac{\Sigma^3\left(\left(\mathbf{k}e^{2\varphi} + 4\right)\cosh\phi - \mathbf{z}\mathbf{k}e^{2\varphi}\sinh\phi\right) + e^{2\varphi}}{4\Sigma^2} \,. \tag{60}$$

Furthermore, the following AdS$_5$ vacuum is also a vev for the scalars $\Sigma, \phi, \varphi$ in this truncation:

$$\begin{aligned}
\varphi &= \frac{1}{2}\log\left(\frac{4}{\sqrt{3\mathbf{z}^2 + 1 - 2\mathbf{k}}}\right)\,, \\
\phi &= \operatorname{arctanh}\left(\frac{1 + \mathbf{k}\sqrt{1 + 3\mathbf{z}^2}}{3\mathbf{z}}\right)\,, \\
\Sigma^3 &= \frac{\sqrt{2\left(3\mathbf{z}^2 - 1 - \mathbf{k}\sqrt{1 + 3\mathbf{z}^2}\right)}}{\mathbf{z}\left(\sqrt{1 + 3\mathbf{z}^2} - 2\mathbf{k}\right)}\,.
\end{aligned} \tag{61}$$

## 4 The 5d truncation on the spindle

In this section we briefly review the geometric construction used to split the five-dimensional background as the warped product AdS$_3 \times \Sigma$, where the space $\Sigma$ is a compact spindle with azimuthal symmetry and conical singularities at the poles. Once introduced the Ansatz on the geometry and on the gauge fields, we present the corresponding BPS equations and Maxwell equations of motion.

We refer the reader to [48] for the original derivation and to [37] for a more detailed analysis made using our conventions.

### 4.1 The Ansatz and Maxwell equations

We begin by considering the AdS$_3 \times \Sigma$ Ansatz made in [48]:[7]

$$\begin{aligned}
ds^2 &= e^{2V(y)}ds^2_{\text{AdS}_3} + f(y)^2 dy^2 + h(y)^2 dz^2 \,, \\
A^{(I)} &= a(y)^{(I)}dz \,,
\end{aligned} \tag{62}$$

where $ds^2_{\text{AdS}_3}$ is the metric on unitary AdS$_3$, while $(y, z)$ are the coordinates on $\Sigma$, which is a compact spindle with an azimuthal symmetry generated by $\partial_z$. A spindle is a weighted

---

[7]We are using the mostly plus signature, as in [37].

projective space $\mathbf{WCP}^1_{[n_N, n_S]}$ with conical deficit angles at the north $(n_N)$ and at the south $(n_S)$ pole, whose geometry is determined by the two co-prime integers $n_N \neq n_S$ that are associated to the deficit angles $2\pi\left(1 - \frac{1}{n_{N,S}}\right)$ at the poles.

The azimuthal coordinate $z$ has periodicity $\Delta z = 2\pi$. The longitudinal coordinate $y$ is compact, bounded by $y_N$ and $y_S$ (with $y_N < y_S$), implying that the function $h(y)$ vanishes at the poles of the spindle. We assume that the scalars $\Sigma, \phi, \varphi$ depend on the $y$ coordinate only, while the hyperscalar $\Xi$ is linear in $z$, i.e. $\Xi = \bar{\bar{\Xi}}z$ (with $\bar{\bar{\Xi}}$ a constant). Following [48], we will use an orthonormal frame to simplify the analysis of the Killing spinor equations and of the equations of motion of the gauge fields:

$$e^a = e^V \bar{e}^a, \qquad e^3 = f\,dy, \qquad e^4 = h\,dz, \tag{63}$$

where $\bar{e}^a$ is an orthonormal frame for AdS$_3$. In this basis, the field strengths read

$$f\,h\,F^{(I)}_{34} = \partial_y a^{(I)}. \tag{64}$$

Given that $\Sigma, \phi, \varphi$ are functions of $y$ only and $\Xi = \bar{\bar{\Xi}}z$, two out of the three gauge equations of motion specified to our Ansatz can be easily integrated, and they can be written in the orthonormal frame as

$$\frac{2e^{3V}}{3\Sigma^2}\Big[(\cosh 2\phi - \mathbf{z}\sinh 2\phi)F^{(1)}_{34} + (\mathbf{z}\cosh 2\phi - \sinh 2\phi)F^{(2)}_{34}\Big] = \mathcal{E}_1, \tag{65}$$

$$\frac{2e^{3V}}{3\Sigma^2}\Big[\mathbf{z}\mathbf{k}\Sigma^6 F^{(0)}_{34} - (\cosh 2\phi + \mathbf{z}\sinh 2\phi)F^{(1)}_{34} + (\mathbf{z}\cosh 2\phi + \sinh 2\phi)F^{(2)}_{34}\Big] = \mathcal{E}_2, \tag{66}$$

$$\partial_y\Big(\frac{1}{3}e^{3V}\Sigma^4 F^{(0)}_{34}\Big) = \frac{1}{4}e^{4\psi + 3V} g\,f\,h^{-1}D_z\Xi, \tag{67}$$

where $\mathcal{E}_1$ and $\mathcal{E}_2$ are constants, and we defined $D_z\Xi \equiv \bar{\bar{\Xi}} + g(a^{(0)} + \mathbf{z}\mathbf{k}a^{(1)} - \mathbf{k}a^{(2)})$.

## 4.2 The BPS equations

To derive the BPS equations for the geometry introduced above, we need to factorize the Killing spinor [48]:

$$\epsilon = \psi \otimes \chi, \tag{68}$$

where $\chi$ is a two-component spinor on the spindle and $\phi$ is a two-component spinor on AdS$_3$ such that

$$\nabla_m\psi = -\frac{\kappa}{2}\Gamma_m\psi, \tag{69}$$

with $\kappa = \pm 1$ depending on the $\mathcal{N} = (2,0)$ or $\mathcal{N} = (0,2)$ supersymmetry chirality of the dual 2d SCFT.

We then decompose the 5d gamma matrices as

$$\gamma^m = \Gamma^m \otimes \sigma^3, \qquad \gamma^3 = \mathbf{I}_2 \otimes \sigma^1, \qquad \gamma^4 = \mathbf{I}_2 \otimes \sigma^2, \tag{70}$$

with $\Gamma^m = (-i\sigma^2, \sigma^3, \sigma^1)$.

The analysis of the BPS equations is similar to the one in appendix C of [37] (or to the original of [48]). Here again the spinor $\chi$ can be written in terms of an auxiliary function $\xi(y)$ as

$$\chi = e^{V/2}e^{isz}\begin{pmatrix} \sin\frac{\xi}{2} \\ \cos\frac{\xi}{2} \end{pmatrix}, \tag{71}$$

with $s$ a constant. Notice that, as expected, the spinor is not constant on the spindle.

In the following we summarize the differential relations coming from the BPS equations

$$\xi' - 2f(gW\cos\xi + \kappa e^{-V}) = 0,$$

$$V' - \frac{2}{3}f\,gW\sin\xi = 0,$$

$$\Sigma' + \frac{2}{3}f\,g\,\Sigma^2\sin\xi\,\partial_\Sigma W = 0,$$

$$\phi' + 2f\,g\sin\xi\,\partial_\phi W = 0,$$

$$\varphi' + \frac{f\,g}{\sin\xi}\partial_\varphi W = 0,$$

$$h' - \frac{2f\,h}{3\sin\xi}(gW(1 + 2\cos^2\xi) + 3\kappa e^{-V}\cot\xi) = 0,$$

(72)

where $W$ is the superpotential defined in (60). Besides the first-order equations, there are also two algebraic constraints that can be derived from the supersymmetry variations

$$\sin\xi(s - Q_z) = -h(gW\cos\xi + \kappa e^{-V}),$$
$$gh\partial_\varphi W\cos\xi = \partial_\varphi Q_z\sin\xi,$$

(73)

where $Q_z$ can be read from the supercovariant derivative $D_\mu\epsilon = \nabla_\mu\epsilon - iQ_\mu\epsilon$ that appears in the gravitino variation and for our model takes the form

$$Q_z = \frac{e^{2\varphi}}{4}D_z\Xi - g\,a^{(2)}.$$

(74)

We can also reduce the differential system by observing that

$$h = ke^V\sin\xi,$$

(75)

where $k$ is an arbitrary constant that needs to be determined. Finally, we can take advantage of the BPS equations to express the field strengths in terms of the scalar fields as

$$F_{34}^{(0)} = \frac{6\kappa e^{-V} + 4gW\cos\xi - 4g\Sigma\,\partial_\Sigma W\cos\xi}{3\Sigma^2},$$

$$F_{34}^{(1)} = -\frac{2\Sigma}{3}\Big[\sinh\phi\big(g\cos\xi(2W + \Sigma\,\partial_\Sigma W) + 3\kappa e^{-V}\big) + 3g\,\partial_\phi W\cos\xi\cosh\phi\Big],$$

$$F_{34}^{(2)} = -\frac{2\Sigma}{3}\Big[\cosh\phi\big(g\cos\xi(2W + \Sigma\,\partial_\Sigma W) + 3\kappa e^{-V}\big) + 3g\,\partial_\phi W\cos\xi\sinh\phi\Big].$$

(76)

# 5 Analysis at the poles

In this section we study the solutions of the BPS equations derived above and we show how to obtain the 2d central charge from the pole analysis. The procedure follows the one originally described in [48] and then applied in [36, 37] for the case of the conifold. We start by summarizing the BPS equations, the constraints and the Maxwell equations. Then we derive the explicit expressions of the conserved charges and the magnetic fluxes. The charge conservation imposes the constraints that allow us to fix the boundary conditions at the poles for the scalars that enter the calculation of the central charge. We then compute the central charge from the Brown-Henneaux formula and discuss its relation with the calculation done on the field theory side.

Before starting our analysis let us stress that, differently from the discussion in [36, 37, 48] we have not found from the pole analysis immediate reasons to exclude the possibility of having solutions in the twist class. We will further comment on this issue in the next section where we provide numeric and analytical solutions of the BPS equations.

## 5.1 Conserved charges and restriction to the poles

From the expressions of the field strengths in (76) we can study the Maxwell equations using the two conserved charges $\mathcal{E}_{1,2}$ in (65) and (66). In order to keep the hyperscalar $\varphi(y)$ finite we require that $\partial_\varphi W|_{N,S} = 0$. This constraint gives rise to

$$\mathbf{k}\,\Sigma|_{N,S}^3 + \frac{1}{\cosh\phi|_{N,S} - \mathbf{z}\sinh\phi|_{N,S}} = 0\,, \tag{77}$$

where $W$ is given in (60). Using (77) and the fact that $\mathcal{E}_1$ and $\mathcal{E}_2$ are conserved we found simpler expressions by working with the following linear combinations

$$Q_1|_{N,S} = \mathcal{E}_1|_{N,S} = \frac{4}{3}e^{2V|_{N,S}}\left(\frac{\kappa(\sinh(\phi|_{N,S}) - \mathbf{z}\cosh(\phi|_{N,S}))}{\Sigma|_{N,S}} - \mathbf{z}g\,e^{V|_{N,S}}\cos(\xi|_{N,S})\right),$$

$$Q_2|_{N,S} = \mathcal{E}_1|_{N,S} - \mathcal{E}_2|_{N,S} = \frac{4\kappa e^{2V|_{N,S}}}{3\Sigma|_{N,S}}\left(2\sinh(\phi|_{N,S}) - \mathbf{z}\mathbf{k}\Sigma|_{N,S}^3\right). \tag{78}$$

At the north and at the south poles we have $k\sin\xi \to 0$. For non-vanishing $k$ this gives $\cos\xi_{N,S} = (-1)^{t_{N,S}}$ with $t_{N,S} = 0$ or $t_{N,S} = 1$. Denoting the poles as $y_{N,S}$ we can work with $y_N \le y \le y_S$. Furthermore,

$$|h'|_{N,S} = |k\sin'\xi|_{N,S} = \frac{1}{n_{N,S}}\,. \tag{79}$$

This relation is due to the metric and to the deficit angles at the poles $2\pi\left(1 - \frac{1}{n_{N,S}}\right)$ where $n_{N,S} > 1$. From the $\mathbf{Z}_2$ symmetry of the BPS equations acting on $h, a^{(I)}, s, Q_z$ and $k$ we can restrict to $h \ge 0$ and $k\sin\xi \ge 0$. We have then $k\sin\xi \ge 0$ and this quantity is vanishing at the poles, with a positive derivative at $y_N$ and a negative one at $y_S$. Formally we introduce two constants, $l_N = 0$ and $l_S = 1$ such that

$$k\sin'\xi|_{N,S} = \frac{(-1)^{l_{N,S}}}{n_{N,S}}\,. \tag{80}$$

Then the cases $(t_N, t_S) = (0,0)$ and $(1,1)$ correspond to the twist while $(t_N, t_S) = (1,0)$ and $(0,1)$ correspond to the anti-twist. Plugging the evaluation of $\cos\xi$ at the poles in (80), we obtain a relation for $\xi'$ at the poles as well. Furthermore, $\xi'$ following from the first BPS equation in (72) in the conformal gauge, can be shown to be proportional to the quantity $(s - Q_z)$ in (73), after plugging in this last the relation (75). It follows that, the quantity $(s - Q_z)$ at the poles becomes

$$s - Q_Z|_{N,S} = \frac{1}{2n_{N,S}}(-1)^{t_{N,S}+l_{N,S}+1}\,. \tag{81}$$

Furthermore, the relation $\partial_\varphi W|_{N,S} = 0$ imposes from the second relation in (73) that $\partial_\varphi Q_z|_{N,S} = 0$. Another assumption (justified a posteriori by the numerical results) is that $\psi|_{N,S} \ne 0$. Such assumption implies also that $D_z\Xi|_{N,S} = 0$.

## 5.2 Fluxes

Here we introduce the magnetic fluxes for the reduction of this truncation on the spindle. This will be necessary in order to find the constant $k$ introduced in (75) in terms of the data of the spindle. First, from the relations (76), we observe that

$$F_{yz}^{(I)} = (a^{(I)})' = \left(\mathcal{I}^{(I)}\right)'\,, \quad \text{with} \quad \mathcal{I}^{(I)} \equiv -ke^V\cos\xi\,h^I\,. \tag{82}$$

At this point we need to define the fluxes starting from (82). Let's start by defining the integer fluxes $p_I$ from the relations

$$\frac{p_I}{n_N n_S} = \frac{1}{2\pi} \int_\Sigma g F^{(I)} = g \mathcal{I}^{(I)}\big|_N^S.$$ (83)

The magnetic charge associated to the $R$-symmetry is

$$-g n_N n_S \mathcal{I}^{(2)}\big|_N^S = \frac{1}{2}\left(n_S(-1)^{t_N} + n_N(-1)^{t_S}\right).$$ (84)

This expression is quantized if $n_S(-1)^{t_N} + n_N(-1)^{t_S}$ is even. Observe also that

$$\mathcal{I}^{(0)} + \mathbf{zk}\mathcal{I}^{(1)} - \mathbf{k}\mathcal{I}^{(2)} = 0,$$ (85)

that implies also that the combination $p_0 + \mathbf{zk}p_1 - \mathbf{k}p_2$ does not give rise to a conserved magnetic flux. The last flux that we need to discuss is the one associated to the flavor symmetry. The integer flavor flux is given by

$$p_F = g n_N n_S \mathcal{I}^{(1)}\big|_N^S.$$ (86)

It is important to observe that the relation $p_0 = \mathbf{k}(\mathbf{z}p_F + p_2) \in \mathbf{Z}$ requires that for $\mathbf{z} \in \mathbf{Q} \setminus \mathbf{Z}$ we have the further constraint $\mathbf{z}p_F \in \mathbf{Z}$.

Furthermore we also found useful to introduce an auxiliary function $\delta$, in terms of which we can rewrite

$$\tanh(\phi) \equiv \frac{1-\delta}{\mathbf{z}},$$ (87)

such that the charges evaluated at the poles simplify to

$$Q_{1N,S} = \frac{\mathbf{k}\delta_{N,S}((-1)^{l_{N,S}} - 2\kappa k n_{N,S}(-1)^{t_{N,S}})^2}{6\mathbf{z}g^2k^3 n_{N,S}^3}$$
$$\times (2\kappa k n_{N,S}(\delta_{N,S}-1)\delta_{N,S} - (-1)^{l_{N,S}-t_{N,S}}((\delta_{N,S}-1)^2 - \mathbf{z}^2)),$$ (88)

$$Q_{2N,S} = \frac{\mathbf{k}\kappa((-1)^{l_{N,S}} - 2\kappa k n_{N,S}(-1)^{t_{N,S}})^2}{3\mathbf{z}g^2k^2 n_{N,S}^2}(\mathbf{z}^2 - 1 + \delta_{N,S}(4 - 3\delta_{N,S})).$$

It follows that we have three equations: the first one is (86), that after the substitution (87) becomes

$$p_F = \frac{(\delta_N - 1)n_S(-1)^{-t_N} + n_N(-1)^{-t_S}(\delta_S - 1) - 2\kappa k n_N n_S(\delta_N - \delta_S)}{2\mathbf{z}},$$

while the other two equations correspond to $Q_1|_N = Q_1|_S$, i.e.

$$\frac{(1 + 2\kappa k n_S(-1)^{t_S})^2}{(1 - 2\kappa k n_N(-1)^{t_N})^2} \cdot \frac{\delta_S n_N^3}{\delta_N n_S^3} \cdot \frac{2\kappa k n_S(-1)^{t_S}(\delta_S - 1)\delta_S + (\delta_S - 1)^2 - \mathbf{z}^2}{2\kappa k n_N(-1)^{t_N}(\delta_N - 1)\delta_N - (\delta_N - 1)^2 + \mathbf{z}^2} = (-1)^{t_S + t_N},$$

and $Q_2|_N = Q_2|_S$, i.e.

$$\frac{n_N^2}{n_S^2} \cdot \frac{\mathbf{z}^2 - 1 + \delta_S(4 - 3\delta_S)}{\mathbf{z}^2 - 1 + \delta_N(4 - 3\delta_N)} \cdot \frac{(1 + 2\kappa k n_S(-1)^{t_S})^2}{(1 - 2\kappa k n_N(-1)^{t_N})^2} = 1,$$ (89)

for the three variables, $k$, $\delta_S$ and $\delta_N$. By solving these three equations we obtain then the boundary conditions to impose for the scalars $V, h, \phi, \Sigma$ in terms of the integers $n_S$, $n_N$ and $p_F$ of the spindle for generic values of the parameters $\mathbf{z} \in \mathbf{Q}$ and $\mathbf{k} = \pm 1$ in both the twist and the anti-twist class. The requirement of reality for these fields imposes further constraints on the allowed values of the integers $n_{S,N}$ and $p_F$. The only field that is not involved in this analysis is the hyperscalar $\varphi$, that we are assuming as non-vanishing at the poles.

## 5.3 Central charge from the pole data

Once the boundary data for $\delta_{N,S}$ and the constant $k$ are specified we can read the central charge of the putative 2d CFT from the pole analysis. The central charge is obtained from the Brown–Henneaux formula [1]

$$c_{2d} = \frac{3R_{AdS_3}}{2G_3} = \frac{3}{2G_5}\Delta z \int_{y_N}^{y_S} e^{V(y)}|f(y)h(y)|dy. \tag{90}$$

The relation

$$e^{V(y)}f(y)h(y) = -\frac{k}{2\kappa}(e^{3V(y)}\cos\xi(y))', \tag{91}$$

implies that the central charge can be computed from the value of the fields at the poles that we have computed above, without specifying the value of the hyperscalar. The consistency of this analysis represents just a necessary condition for the existence of a solution. Nevertheless, when a solution exists, the central charge computed here is the correct one.

In the conformal gauge $f = e^V$ the integrand in (90) is $e^{V(y)}|h(y)|$, where we remove the absolute value here and consider $h(y) > 0$ thanks to the symmetries of the BPS equations as discussed above. The central charge becomes $c_{2d} = c_S - c_N$ where

$$c_{N,S} = \frac{3\pi \mathbf{k}\delta_{N,S}}{2\mathbf{z}^2 g^3 G_5 \kappa k^2}\left(\kappa k - \frac{(-1)^{l_{N,S}-t_{N,S}}}{2n_{N,S}}\right)^3 ((\delta_{N,S}-1)^2 - \mathbf{z}^2). \tag{92}$$

The central charge in the case of the anti-twist, splitting numerator and denominator for ease of readability, is given by

$$\text{Numerator} = 3\pi \mathbf{k}\kappa(4p_F^2 - (n_S - n_N)^2)(2\mathbf{z}p_F(-1)^{t_N} - n_N + n_S)$$
$$\times (n_S - n_N)\Big(16\mathbf{z}p_F(-1)^{t_N} + (\mathbf{z}^2 + 3)(n_S - n_N) + 4(3\mathbf{z}^2 + 1)p_F^2\Big), \tag{93}$$
$$\text{Denominator} = 4g^3 G_5 n_N n_S\Big(8\mathbf{z}p_F(-1)^{t_N}(n_S - n_N)(3n_N^2 + 2n_N n_S + 3n_S^2 - 4p_F^2)$$
$$+ 16p_F^2 n_N n_S + 4(n_S - n_N)(n_S^3 - n_N^3)$$
$$+ \mathbf{z}^2\Big(24p_F^2(n_N^2 + n_S^2) - 48p_F^4 + (n_S^2 - n_N^2)^2\Big)\Big), \tag{94}$$

while the central charge in the case of the twist is given by

$$\text{Numerator} = 3\pi \mathbf{k}\kappa(4p_F^2 - (n_S - n_N)^2)(2\mathbf{z}p_F(-1)^{t_N} - n_N + n_S)$$
$$\times \Big((n_N + n_S)(16\mathbf{z}p_F(-1)^{t_N} + (\mathbf{z}^2 + 3)(n_N + n_S)) + 4(3\mathbf{z}^2 + 1)p_F^2\Big), \tag{95}$$
$$\text{Denominator} = 4g^3 G_5 n_N n_S\Big(8\mathbf{z}p_F(-1)^{t_N}(n_N + n_S)(3n_N^2 - 2n_N n_S + 3n_S^2 - 4p_F^2)$$
$$- 16p_F^2 n_N n_S + 4(n_N + n_S)(n_N^3 + n_S^3)$$
$$+ \mathbf{z}^2\Big(24p_F^2(n_N^2 + n_S^2) - 48p_F^4 + (n_S^2 - n_N^2)^2\Big)\Big). \tag{96}$$

The five dimensional Newton constant can be read from the holographic dictionary. Indeed from the general relation $a_{4d} = \frac{\pi R_{AdS_5}^3}{8G_5}$ and from the explicit values of the central charge and of the AdS$_5$ radius, given by

$$a_{4d} = \frac{(g-1)\Big((1-9\mathbf{z}^2)\mathbf{k} + (3\mathbf{z}^2+1)^{3/2}\Big)}{48\mathbf{k}\mathbf{z}^2}, \qquad R_{AdS_5}^3 = \frac{(1-9\mathbf{z}^2)\mathbf{k} + (3\mathbf{z}^2+1)^{3/2}}{4\mathbf{z}^2}, \tag{97}$$

we can extract $G_5 = \frac{3\pi \mathbf{k}}{2(g-1)}$. Substituting this expression in the 2d central charge computed above we can then recover the result obtained from the field theory calculation in Section 2.2.

Some comments are in order. First we have checked in many cases if the various constraints, imposed by the quantization of the fluxes, by the reality condition on the scalars and by the positivity of the central charge, are enough to exclude the existence of some solutions. While in many cases the answer is affirmative, we have not been able to exclude whole families of solutions. In general there are four main families of possible solutions, identified by the value of $\mathbf{k} = \pm 1$ and by the fact that they can be in the twist or in the anti-twist class. Anyway, anticipating the results of next section, we have found solutions only in the anti-twist class for $\mathbf{k} = -1$.

## 6 The solution

In this section we obtain the AdS$_3 \times \Sigma$ solution for the model discussed above. We separate the analysis in two parts. In the first part we discuss the analytic solution for the universal truncation. This corresponds to a further truncation of the model to the graviton sector. In this case we found the explicit solution corresponding to the general one found in [13, 15]. Similarly to the cases discussed in [36,37,48] in presence of hypermultiplets, here we found an analytic solution only in the anti-twist class. Furthermore, we have found such solution only for $\mathbf{k} = -1$. We have also checked that the 2d central charge matches the general expectation [13]

$$c_{2d} = \frac{4}{3} \frac{a_{4d}(n_S - n_N)^3}{n_N n_S (n_N^2 + n_N n_S + n_S^2)} \,. \tag{98}$$

In the second part of this section we study the solution turning on a generic flux $p_F$. In this case we have obtained the solution numerically. Again we found solutions only in the anti-twist class for $\mathbf{k} = -1$ and for generic values of $\mathbf{z}$.

### 6.1 Analytic solution for the graviton sector

Here we study the AdS$_3 \times \Sigma$ solution by restricting to the graviton sector. It will turn out that the solution is exactly the same as the one studied in the original Spindle paper [13]. This is consistent with similar results obtained in other 5d truncations in presence of hypermultiplets [36, 37, 48]. This requires to fix $A^{(1)} + \epsilon_{4d}^* A^{(2)} = 0$ (with $\epsilon_{4d}^*$ defined in (6)) and identifying $A^{(R)} = -A^{(2)}$. This further fixes $2p_F = \epsilon_{4d}^*(n_S - n_N)$. We have found a solution in this case for the anti-twist class and $\mathbf{k} = -1$ by fixing the scalars $\Sigma(y)$, $\phi(y)$ and $\varphi(y)$ at their AdS$_5$ value (61). Observe that $\phi_{N,S} = \phi_{AdS_5}$ and $\Sigma_{N,S} = \Sigma_{AdS_5}$ when $p_F = \epsilon_{4d}^*(n_S - n_N)/2$.

Before continuing the discussion a comment is in order. The choice of $p_F$ that allows to study the universal twist is, for generic values of $\mathbf{z}$, in contrast with the requirement that $\mathbf{z}p_F$ is an integer. The only cases that are allowed correspond to the ones that give rise to a rational exact $R$-symmetry. In these cases a solution exists when (the even quantity) $n_S - n_N$ gives rise to an integer $\mathbf{z}p_F$. This analysis restricts the possible truncations to the graviton sector that can be placed on the spindle. This is the counterpart of the field theory argument that we made after formula (30). The discussion fits with similar ones appeared in the literature of the spindle (see for example footnote 20 of [16] for an analogous behavior in the case of toric SE$_5$). Having this caveat in mind, the scalar functions $V(y)$, $f(y)$ and $h(y)$ in (62) are

$$e^{V(y)} = \frac{\sqrt{y}}{W}, \qquad f(y) = \frac{3}{2W}\sqrt{\frac{y}{q(y)}}, \qquad h(y) = \frac{c_0 \sqrt{q(y)}}{4Wy}, \tag{99}$$

while the gauge field is

$$A^{(R)} = \left( \frac{c_0 \kappa (a-y)}{4y} - s \right) dz \, . \tag{100}$$

We also found that

$$\sin \xi(y) = \frac{\sqrt{q(y)}}{2y^{3/2}} \, , \qquad \cos \xi(y) = \frac{\kappa(3y-a)}{2y^{3/2}} \, , \tag{101}$$

with

$$q(y) = 4y^3 - 9y^2 + 6ay - a^2 \, . \tag{102}$$

The constants $a$ and $c_0$ are obtained from the solutions of the BPS equations at the poles. We found

$$c_0 = \frac{2\left(n_N^2 + n_N n_S + n_S^2\right)}{3 n_N n_S \left(n_N + n_S\right)} \, , \tag{103}$$

while the constant $a$ is

$$a = \frac{(n_N - n_S)^2 (2n_N + n_S)^2 (n_N + 2n_S)^2}{4\left(n_N n_S + n_N^2 + n_S^2\right)^3} \, . \tag{104}$$

From here it follows that

$$y_N = \frac{\left(-2n_N^2 + n_N n_S + n_S^2\right)^2}{4\left(n_N^2 + n_N n_S + n_S^2\right)^2} \, , \qquad y_S = \frac{(n_N - n_S)^2 (n_N + 2n_S)^2}{4\left(n_N^2 + n_N n_S + n_S^2\right)^2} \, . \tag{105}$$

The central charge becomes

$$c_{2d} = \frac{9\pi (n_S - n_N)^3}{16 G_5 W_{\text{crit}}^3 n_N n_S \left(n_N^2 + n_N n_S + n_S^2\right)} \, . \tag{106}$$

Using then $a_{4d} = \frac{\pi R_{AdS_5}^3}{8G_5}$ and $R_{AdS_5} = \frac{3}{2W_{\text{crit}}}$ we arrive at the expected universal result (98). Observe that $W_{\text{crit}}$ can be consistently found from the relation (97).

Before turning on a generic value for the flavor flux $p_F$ and studying the solution numerically, a comment is in order. We have so far referred to the solution with $p_F$ set to $\epsilon_{4d}^*(n_S-n_N)/2$ as "universal" solution. Such terminology refers to the fact that the truncation is restricted the "pure" gravity sector, indeed recovering the AdS$_5$ vacuum. On the field theory side the constraint on $p_F$ indeed reflects on the ones on $p$ and $q$ that set the exact $R$-symmetry to be rational, as discussed at the end of subsection 2.2.

## 6.2 Numerical solution for generic $p_F$

Here we look for more generic solutions of the BPS equations interpolating among the poles of the spindle. From the analysis above we have observed that the analytic solutions with $p_F = \epsilon_{4d}^*(n_S - n_N)/2$ are in the anti-twist class with $\mathbf{k} = -1$. Here we search for numerical solutions for a generic integer $\mathbf{z}p_F$. We have scanned over large regions of parameters and again we have only found solutions with $\mathbf{k} = -1$ in the anti-twist class.

The solutions are found along the lines of the analysis of [36, 37, 48]. First we specify the values of $\mathbf{z}$, $n_S$, $n_N$ and $p_F$. Then we fix the initial conditions imposed by the analysis at the poles. In this way we are left with one unknown initial condition for the hyperscalar $\varphi$. Finding the initial condition of $\varphi$ corresponds to find the solution for the BPS equations on the spindle. There is just (up to the numerical approximation) a single value $\varphi_S$ (here we are fixing the south pole at $y_S = 0$) that allows to integrate the BPS equation giving rise to a finite

Table 1: Some numerical solutions found in our analysis for various, consistent, values of deficit angles $n_{N,S}$, flavor flux $p_F$ and geometry $\mathbf{z}$. The boundary values for the field $\varphi$ are found so to give rise to a finite Spindle in the $y$ direction.

| $n_S$ | $n_N$ | $p_F$ | $\mathbf{z}$ | $\varphi_S$ | $\varphi_N$ | $\Delta y$ |
|---|---|---|---|---|---|---|
| 1 | 3 | 0 | 2 | -0.285076 | -0.274493 | 1.83241 |
| 1 | 7 | -1 | 2 | -0.172372 | -0.170589 | 2.39707 |
| 1 | 3 | 0 | 3 | -0.555814 | -0.542721 | 1.82303 |
| 1 | 5 | -1 | 3 | -0.300428 | -0.300346 | 2.16012 |
| 1 | 9 | 3 | $\frac{1}{3}$ | 0.463989 | 0.363277 | 2.57446 |
| 1 | 5 | 0 | $\frac{1}{3}$ | 0.126802 | 0.124497 | 2.16392 |
| 1 | 7 | 2 | $\frac{1}{2}$ | 0.484886 | 0.347516 | 2.3322 |
| 3 | 7 | 0 | $\frac{1}{2}$ | 0.104192 | 0.103447 | 1.74866 |

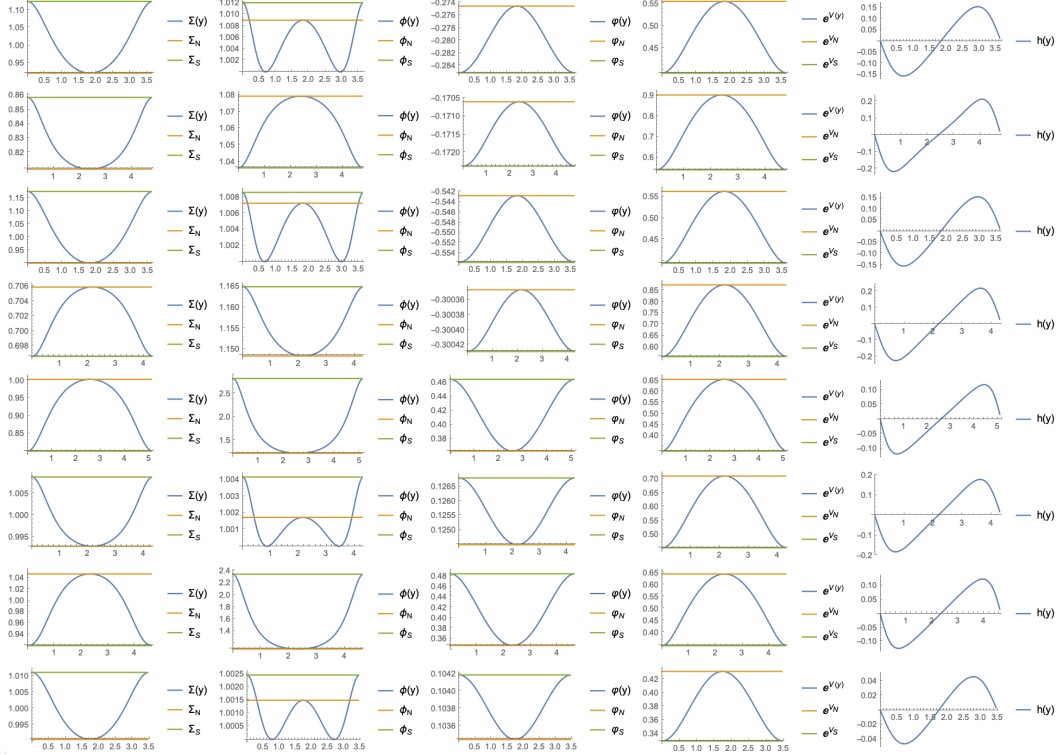

Figure 1: Numerical solutions for the scalar fields $\Sigma(y), \phi(y), \varphi(y)$ and the scalar functions $e^V(y)$ and $h(y)$ interpolating between $y = y_S = 0$ and $y = 2(y_N - y_S)$. The values of $n_S, n_N, p_F$ and $\mathbf{z}$ are the ones fixed in (1). The values of the fields at $y_S$ are the green lines and at $y_N$ are the orange ones. From left to right, the plots follow the same order of table 1.

spindle in the $y$ direction. Once this value is found a good sanity check consists of running the numerics until $2\Delta y$, that corresponds to solve the BPS equations from the north to the south pole as well. We have scanned over various values of the parameters and we present some of the solutions that we found in table 1.

In each case we have fixed $\mathbf{k} = -1$ and chosen $\kappa = 1$ (corresponding to the choice $n_N > n_S$). The explicit solutions are plot in Figure 1. Observe that the solutions for the cases at $p_F = 0$ do not correspond to the universal twist (at least for $\mathbf{z} \neq 0$). The cases at $p_F = 0$ correspond to a twist along a trial $R$-symmetry, obtained from a linear combination (with irrational coefficients) of the (irrational) exact $R$-symmetry and the flavor symmetry.

# 7 Conclusions

In this paper we studied the reduction of the consistent truncations found in [43] on the spindle. These truncations are associated to M5 branes wrapping holomorphic curves in a $CY_3$ and the dual field theories have been obtained in [40, 41]. Using these results we matched the 2d central charge obtained from the field theoretical analysis with the one predicted in gauged supergravity from the analysis at the poles of the spindle. We have also studied the full solution, showing its existence for consistent choices of the parameters, analytically for the universal anti-twist and numerically after including the magnetic charge of the flavor symmetry.

There are many interesting aspects that we did not investigate. A first open question regards the uplift of our solutions to 7d and 11d supergravity. An interesting limit corresponds to set $\mathbf{z} = \pm 1$ and consider $p_F = 2\mathbf{z}\big(q - \frac{1}{4}(n_S - n_N)\big)$. In this case we reproduce the results obtained in [17] for the $\mathcal{N} = 2$ Maldacena-Nuñez theory. Observe that the matching works when $p_F$ and $(n_S - n_N)/2$ have the same parity.

Another open question regards the existence of solutions for $\mathbf{k} = 1$ and $|\mathbf{z}| > 1$ in both the twist and the anti-twist class and for $\mathbf{k} = -1$ in the twist class. Even if we have not been able to exclude these possibilities (for generic values of $\mathbf{z}$) we have not found any solution of this type neither in the analytical nor in the numerical analysis carried out in section 6. Nevertheless we observe that by choosing $\mathbf{z} = 0$ we can simplify the problem (for $\mathbf{k} = -1$) and we obtain results similar to the one studied in [36, 37, 48]. This limit corresponds to the $\mathcal{N} = 1$ Maldacena-Nuñez theory and in this case the pole analysis completely excludes the existence of solutions in the twist class. The reason is that in this case we can impose further reality constraints on the conserved charges against the existence of such solutions.

Our analysis has been performed at leading order in $N$, i.e. the central charge here is scales with $N^3$. There is a subleading contribution of order $N$, proportional to the gravitational anomaly of the SCFT, that one could compute from the field theoretical side. It would be interesting to match this contribution from the holographic analysis. A similar calculation was carried out for the case of the topological twist in [53].

It would also be interesting to consider M5 branes wrapping other geometries. For example by considering a disc, an holographic dual of an $\mathcal{N} = 2$ SCFT of AD type was proposed in [54–56] (see also [27]) As then observed in [57, 58] indeed the disc and spindle geometries are different global completions of the same local solution.

Finally, it would be possible to study the models discussed here from the 11d perspective along the lines of the recent discussions of [59–62] from the theory of equivariant localization.

## Acknowledgments

S.M. wants to thank the Université Libre de Bruxelles (ULB) for the warm hospitality.

**Funding information** The work of A.A., A.S. and D.M. has been supported in part by the Italian Ministero dell'Istruzione, Università e Ricerca (MIUR), in part by Istituto Nazionale di Fisica Nucleare (INFN) through the "Gauge Theories, Strings, Supergravity" (GSS) research project and in part by MIUR-PRIN contract 2017CC72MK-003. The work of N.P. is supported by the Israel Science Foundation (grant No. 741/20) and by the German Research Foundation through a German-Israeli Project Cooperation (DIP) grant "Holography and the Swampland". The work of S.M. is supported by "Fondazione Angelo Della Riccia".

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
