# Peer review of "BBBW on the spindle"

_SciPost Physics, doi:SciPost Phys. 17, 154 (2024)_

## Round 1 · Referee Report · Anonymous (Referee 2) · 2024-1-22

Report

The gravity dual of 4d N=1 SCFTs of class S are AdS_5 x Riemann surface solutions in seven-dimensional gauged supergravity. In this paper, the authors construct new supersymmetric solutions of AdS_3 x spindle which are obtained by further compactifying AdS_5 on AdS_3 x spindle. Thus the new solutions are dual to 4d N=1 SCFTs compactified on a spindle.

Although previously AdS_3 x spindle solutions have been constructed in minimal gauged supergravity and STU models in five dimensions, the new solutions generalize the previous ones by introducing non-trivial hypermultiplets dual to flavor charges in 4d N=1 SCFTs.

The authors also perform anomaly polynomial calculations and the results match the holographic central charge obtained from the supergravity solutions.

The paper is clearly written and well organized. The method of constructing solutions is not new, but it was applied to a new and interesting problem. It would be nice to include a more comprehensive review of the supergravity model, e.g. the Lagrangian of the truncation being studied and also the combinations of the gauge fields in the model, but we leave the choice to the authors. There is a small typo, “fields strengths”, above (5.1) which needs to be corrected to be “field strengths”. The paper contains an interesting addition to the recent development of the field and I suggest publication in SciPost Physics.

---

## Round 1 · Referee Report · Anonymous (Referee 3) · 2024-2-6

Report

In this paper the authors consider spindle compactifications of BBBW theories arising from M5-branes wrapping complex curves inside a CY 3-fold and exhibit a non-trivial check of the AdS/CFT correspondence by holographically matching the central charge. The introduction states clearly the scope of the paper and is complemented with a complete set of references to the relevant literature. To reach their goal, the authors employ the techniques firstly illustrated in [48] with suitable adaptations to the case considered. The results obtained are certainly interesting and successfully expand the spindle literature in a novel context.

However, in the manuscript I have encountered a series of issues of various degree of significance and I would like the authors to address them before recommending the publication.

Conceptual: - A consistency check of the result for the central charge in the anti-twist case (see e.g. (2.24) and (2.28)) can be achieved by reproducing the one obtained in [17] when $z=\pm 1$. The authors correctly point this out in the conclusions but unfortunately I was unable to reproduce their claim. I would like them to clarify how this limit works and include the explanation in the main text too. - At many stages in the paper the authors talk about the universal twist and how it is consistent only when the R-symmetry is rational, and then proceed to impose a particular constraint on the quantized magnetic flux. On the other hand, the definition of universal twist I am aware of consists in imposing that the magnetic flux through the spindle of the background field associated to a U(1) symmetry is proportional to the corresponding R-charge. It is not obvious to me that the two definitions coincide. Furthermore, my understanding is that the R-symmetry being rational is a general requirement of these kind of solutions (even outside the universal twist), and indeed it could be achieved even without assuming your constraint on $p_F$ (there are choices for $z$ such that $\epsilon^*$ is rational). - In section 6.1 the authors present an analytic solution for the universal twist. However, at first sight this solution seems to be exactly the same as the one illustrated in the original spindle paper [13]. In turn, since [17] it has been known that this solution can uplift to 11d on the Maldacena-Nuñez solution, which is what the authors seem to do in this section given that the parameter $z$ does not appear anywhere. Of course I might have misunderstood, so I would like the authors to clarify what is the novelty of the solution presented in this section.

Exposition: - Around equations (2.2) and (2.3) I suggest the authors comment on the positivity of $p$ and $q$ and the consequent constraints on z. - At the end of section 2.1 there are a few issues: in (2.5) $R^*$ should appear instead of $R$ (and therefore I suggest the authors introduce $R^*$ before); the parameter $k$ should be 1/2 times the Ricci curvature according to [43] (see also below (3.2)); in (2.8) the authors write the $N$ order for one the coefficients because the $N^3$ order vanishes but this cannot be obtained from (2.7) as the latter captures only the $N^3$ order, so I advice to remove the $N$ order or clarify how it is obtained. - Throughout section 2 there are two different quantities both called $\epsilon$ (the mixing parameters in 4d and in 2d) and this could be confusing. I suggest to use different notations. - Unfortunately I was not able to reproduce the central charge (2.24) by extremizing (2.19) as described (not even in easier limits). I would like the authors to double check their formulas for potential typos. - In equation (2.24) the authors introduce the quantity $a_{4d}$. Though the name is reminiscent of a central charge in 4d, they do not explain if this is the correct interpretation and why. Moreover, later in the paper (c.f. (5.22)) a different quantity with the same name appears. Please clarify what is $a_{4d}$ and fix the clash of notation. - In the paragraph below (2.27) an interesting limit is considered to make contact with the results of [6]. However, it seems to me that this matching requires to set $p_F = 0$ too, and this is not mentioned in the paper. Moreover, I would like to direct the attention of the authors on the fact that in this limit $\epsilon =0$ which means there is no mixing with the U(1) isometry of the spindle. I think it could be interesting to comment on this. - In sections 4 and 5 the authors illustrate a quite technical procedure which is needed to find the gravitational central charge. In general at many points of the derivation I found some non-trivial statements that I was not so sure where they came from, as well as some new quantities appearing without a definition. To name a few (not all) examples: it is not clear what is $\xi$ in (4.10), as well as $\delta$ in (5.11); (5.1), (5.5), (5.6) are all non-trivial consequences of other equations; above (5.19) the authors mention a "conformal gauge" but they do not say what they mean by that. I am aware that most of these confusions could be solved by looking at the derivation in [48] but I think the paper should be fairly self-contained and, although it is not necessary to spell every detail of this, I suggest the authors to at least reference the relevant equations from which each statement is derived. - I think it would be sensible to cite [1] before (5.17). - In the introduction to section 6 I would suggest the authors to explain where (6.1) comes from and/or cite the relevant papers. - Around equation (6.9) I would advice the authors to define what $W_{\text{crit}}$ is. - In the first paragraph of section 6.2 the authors claim that the solutions presented in 6.1 are the "only possible analytic solutions". I think the statement is a bit too strong and perhaps they should consider a different phrasing. - In the third to last paragraph of the conclusions the authors claim to have computed the subleading correction of order $N$ to the central charge from the field theory side but they did not provide any evidence of this in the paper. I would ask the authors to clarify this point and remove/rephrase this sentence.

Typographical/layout: - I would like to point out that all the results for the anti-twist are obtained from those for the twist by sending $n_S \to - n_S$. Therefore in many instances there is no need to write separate equations, especially given how cumbersome they are. I advice you to remove equations (2.28), (2.29), (2.35), (2.36), (2.37) and (5.20) and encorporate these results together with the twist case or make the comment above. - I would advice the authors to modify the layout of (5.21) to something along the lines of the field theory result to improve readability. - In figure 1 at pag. 28 it is not explicitly stated which plot corresponds to which set of numerical parameters. I would ask the authors to add this information.

---

## Round 2 · Referee Report · Anonymous (Referee 3) · 2024-7-25

Report

I thank the authors for addressing all the points I raised in my previous report. I am satisfied with the modifications submitted in the second version of the paper.

I am happy to recommend publication, although I would encourage the authors to further clarify within the paper the derivation and interpretation of the universal twist results - particularly in section 2 since in the current version not much has changed on that respect.

Recommendation

Publish (meets expectations and criteria for this Journal)

---

## Round 2 · List of Changes

We thank the referees for their suggestions. We corrected the typos that found by the first referee and implemented the suggestions from the second referee. It follows a detailed list of changes.

Conceptual
- First bullet: We are grateful to the referee for finding the typos in the coefficient of the anomaly polynomial. We corrected them and checked that they give the right results
- Second bullet: We are extremely grateful to the referee for pointing out a potential misleading interpretation of our results. The deep reason why we referred to the analytic solution in (6.1) as the “universal twist” was hidden, in the previous version of the paper, in the discussion after formula (2.30). Indeed, the requirement of rationality of the exact R-symmetry imposes some constraints that boil down to a restricted set of choices for the parameter p,q that are exactly the ones that the referee refers to in the report. In this way, indeed, restricting to the gravity sector on the holographic dual side corresponds to a twist along the exact R-symmetry with the values of p,q, fixed consistently. This is the usual universal twist and as a check we can see that indeed the AdS5 vacuum is recovered in our construction.
- Third bullet: We quoted reference [13] before formula (6.1) in order to clarify the structure of the solution presented in this section. Furthermore, the dependence from the parameter \mathbb{z} of the central charge is now clarified by the comment after formula (6.9) in terms of W_crit, or equivalently R_AdS5 

Exposition
- First bullet: We added a comment after equation (23).
- Second bullet: We corrected the typos pointed out by the referee and adjusted the formulas. The anomaly in R2F was removed as suggested by the referee.
- Third bullet: we choose to differentiate between the two epsilons by adding a subscript 2d,4d which should make clearer the discussion.
- Fourth bullet: As commented on the first bullet of the conceptual errors, there were some typos in the anomaly polynomial which we now have fixed
- Fifth bullet: as pointed out, the a_{4d} was not needed and we removed it.
- Sixth bullet: as required by the referee, we added a comment on the limit. Indeed, in the sphere limit the U(1) symmetry enhances to SU(2) and therefore cannot mix with the R-symmetry anymore. As a consequence the mixing parameter goes to zero, as expected.
- Seventh bullet: we have inserted the missing definitions. In this way the paper should be self-contained, we thank the referee for pointing out this lack of clarity.
- Eighth bullet: We have inserted the proper citation.
- Ninth bullet: We have cited the paper where (6.1) was originally derived.
- Tenth bullet: We have added a reference to the definition of W_{crit} given in an earlier section of the paper
- Eleventh bullet: as pointed out by the referee, the sentence was misleading and we changed it.
- Twelfth bullet: As noted by the referee, we did not compute the finite-N contributions, so we deleted the highlighted sentence.

Typographic
- First bullet: We think that the explicit formulas are better suited rather than a simple comment about the n_S - >-n_S
- Second bullet: The layout was modified to one similar to the field theory result
- Third bullet: We added a comment in the caption of the figure to highlight to which parameters of Table 1 the plots are referenced to.

---

## Round 3 · Author Response

We are grateful to the referee for the comments, and we commented properly in the body of the paper.

---

## Round 3 · List of Changes

We added footnote n. 4 on page 11 in order to clarify the relation between the universal twist and the procedure spelled out in the paper.

---

## Editorial Decision

published